# TimeSeg: An Information-Theoretic Segment-Wise Explainer for Time-Series Predictions

**Hwijin Kim**
Korea University, Korea
`hwijin0160@gmail.com`

**Jaeho Kim**
Korea University, Korea
`kjh3690@korea.ac.kr`

**Changhee Lee**[*]
Korea University, Korea
`changheelee@korea.ac.kr`

## Abstract

Explaining predictions of black-box time-series models remains a challenging problem due to the dynamically evolving patterns within individual sequences and their complex temporal dependencies. Unfortunately, existing explanation methods largely focus on point-wise explanations, which fail to capture broader temporal context, while methods that attempt to highlight interpretable temporal patterns (*e.g.*, achieved by incorporating a regularizer or fixed-length patches) often lack principled definitions of meaningful segments. This limitation frequently leads to fragmented and confusing explanations for end users. As such, the notion of segment-wise explanations has remained underexplored, with little consensus on what constitutes an *interpretable* segment or how such segments should be identified. To bridge this gap, we define segment-wise explanation for black-box time-series models as the task of selecting contiguous subsequences that maximize their joint mutual information with the target prediction. Building on this formulation, we propose TimeSeg, a novel information-theoretic framework that employs reinforcement learning to sequentially identify predictive temporal segments at a per-instance level. By doing so, TimeSeg produces segment-wise explanations that capture holistic temporal patterns rather than fragmented points, providing class-predictive patterns in a human-interpretable manner. Extensive experiments on both synthetic and real-world datasets demonstrate that TimeSeg produces more coherent and human-understandable explanations, while achieving performance that matches or surpasses existing methods on downstream tasks using the identified segments. Codes are available here.

## 1 Introduction

Explaining time-series models is particularly challenging due to their contiguous, interconnected structure and complex temporal dynamics. These challenges hinder the adoption of time-series models in high-stakes domains, such as healthcare, finance, and manufacturing, where decisions must remain interpretable while operating under strict black-box constraints that prohibit access to model internals (Lipton, 2018; Doshi-Velez & Kim, 2017). In light of this, effective explanations for time series must meet two key criteria simultaneously: (i) accurately identifying specific regions of the time series that drive model predictions and (ii) presenting these regions as human-interpretable temporal segments (Küsters et al., 2020; Queen et al., 2023).

Early efforts to explain time-series models adapted general-purpose explanation techniques (Sundararajan et al., 2017; Lundberg & Lee, 2017; Ribeiro et al., 2016), which treat each time point independently. These point-wise methods fail to capture the temporal dependencies critical for interpreting time-series predictions (Ismail et al., 2020; Leung et al.). To address this, subsequent works have introduced temporal contiguous constraints to encourage contiguous patterns in point-wise explanations (Crabbé & Van Der Schaar, 2021; Liu et al., 2024) or employed patch-wise explanations that explicitly model temporal segments using fixed-length patches (Sivill & Flach, 2022). However, these approaches face trade-offs: point-wise methods excel at precise localization but struggle to produce contiguous temporal patterns, while patch-wise methods enhance interpretability through segment-wise explanations at the cost of reduced localization accuracy due to fixed patch sizes.

---

[*]Corresponding author.



**Figure 1: A Comparison of Generated Explanations for the MIT-ECG Dataset.** TimeSeg produces compact, variable-length *segment-wise explanations* that align well with ground truth, while prior point-wise or patch-wise explanations are fragmented and misaligned.

To bridge this gap, we propose TimeSeg, a post-hoc segment-wise explainer for time series that operates under strict black-box assumptions and *dynamically selects a set of contiguous, variable-length segments within an individual sequence.* TimeSeg formulates segment selection as a sequential decision problem, maximizing the variational lower bound of the joint mutual information (MI) between the model's prediction and the selected segments, while incorporating a sparsity penalty to encourage compact, non-overlapping segments that enhance human interpretability. As illustrated in Figure 1, our approach departs from existing approaches that fall short in different ways: *point-wise* methods (*e.g.*, WinIT (Leung et al.), TimeX++ (Liu et al., 2024)) produce saliency regions that are fragmented by temporal gaps, making them difficult to interpret as coherent temporal patterns, whereas *patch-wise* methods (*e.g.*, LIMESegment (Sivill & Flach, 2022)) enforce contiguous saliency regions but lack precise and variable-length localization. In contrast, TimeSeg identifies compact segments whose boundaries closely align with the ground-truth explanatory region, providing segment-wise explanations that are both temporally coherent and precisely localized.

Our key contributions are summarized as follows:

- We formally define *segment-wise explanation* for time series as an information-theoretic optimization problem that seeks to maximize the MI between the selected segments and the black-box model's predictions while keeping the explanation structurally simple (*i.e.*, concise and non-redundant). To make this problem tractable, we reformulate the joint MI maximization into a sequential process by decomposing it into conditional MI (CMI) terms via the chain rule.
- We cast the sequential decision process of maximizing the CMI terms as a reinforcement learning (RL) problem. In this framework, the explainer acts as an agent that interacts with the black-box model, learning a policy for segment selection that maximizes the expected cumulative reward based on the CMI terms.
- We address a *strict* black-box setting – a more practical scenario where access is restricted to only the model's inputs and outputs – and demonstrate that our method achieves performance comparable or superior to state-of-the-art explainers that require internal access (*e.g.*, gradients or embeddings) across a variety of synthetic and real-world datasets.

## 2 RELATED WORK

Initial efforts in explaining time-series models largely focused on adapting general-purpose explanation methods – such as gradient-based methods (Sundararajan et al., 2017; Shrikumar et al., 2017), perturbation-based methods (Lundberg & Lee, 2017), and surrogate models (Ribeiro et al., 2016) – to time-series data. However, these methods often treat features at individual time points as independent, failing to capture the temporal dependencies that are crucial for interpreting time-series predictions (Ismail et al., 2020; Leung et al.).

**Point-wise Explanation.** To address these limitations, point-wise attribution methods explicitly model temporal relationships when assigning importance to individual time points. Specifically, Dynamask (Crabbé & Van Der Schaar, 2021) proposes a *memoryless* optimization approach for learning dynamic saliency masks. It models the selection of each time point as an independent Bernoulli random variable to relax the combinatorial search problem, and incorporates regularization to encourage the resulting mask to promote temporal smoothness and sparsity. Building on this idea, ExtrMask (Enguehard, 2023) jointly learns both the mask and the perturbation process to better capture temporal dependencies in model predictions. Similarly, FIT (Tonekaboni et al., 2020) estimates point-wise importance by measuring shifts in the predictive distribution via KL-divergence,

while WinIT (Leung et al.) aggregates these shifts over predefined windows to capture local temporal patterns. More recently, TimeX (Queen et al., 2023) and TimeX++ (Liu et al., 2024) employ an *amortized* surrogate explainer to produce point-wise explanations, leveraging a model-behavior consistency objective and straight-through estimators (STE) (Jang et al., 2017) to enable backpropagation through discrete sampling of independent Bernoulli masks. Despite these advances, point-wise methods remain fundamentally constrained by their design: even with regularization to encourage contiguity, they often yield fragmented explanations that highlight isolated time points rather than the coherent temporal segments essential for better intuitive interpretability.

**Segment-wise Explanation.** Some methods have attempted to generate segment-wise explanations. LIMESegment (Sivill & Flach, 2022) adapts LIME (Ribeiro et al., 2016) to a temporal setting by partitioning time series into fixed-size, non-adaptive patches. More recently, SpectralX (Chung et al., 2024) provides frequency-aware explanations by applying perturbations in the frequency domain before converting back to the time domain. However, a fundamental limitation of these approaches is their inability to identify variable-length segments that adapt to the unique temporal dynamics of each time series sample, often leading to misaligned or fragmented explanations (see Figure 1).

In contrast, we introduce a novel framework for selecting *important temporal segments* rather than independent time points. While the success of continuous relaxation in point-wise methods might suggest a natural extension to segments, this adaptation is far from straightforward. Point-wise methods remain tractable because they model each time point as an independent binary variable, but this assumption breaks down for segments, where consecutive time points must be considered jointly. Defining a tractable probability distribution over all possible segments is inherently challenging, as consecutive time points exhibit strong dependencies, and the selection of multiple segments introduces complex combinatorial interdependencies. To overcome these obstacles, we formally define what constitutes a meaningful time-series segment and propose an RL–based framework that sequentially selects segments on a per-instance basis.

## 3 PROBLEM FORMULATION

In this section, we begin by formally defining segment-wise explanations for a *black-box* time-series classifier, followed by a discussion of the key challenges in deriving such explanations.

**Notation.** We consider a pre-trained, *black-box* time-series classifier, $g_\theta : \mathbb{R}^T \to \mathbb{R}^C$, which takes a univariate time series $\mathbf{x} = (x_1, \ldots, x_T) \in \mathbb{R}^T$ and outputs predictions for the target label $y \in [C]$ with $C$ classes. Here, $g_\theta$ is a *strict* black-box such that we can evaluate the model output $g_\theta(\mathbf{x})$ for any given instance $\mathbf{x}$, but we do not have any knowledge of the internal model states, such as parameters ($\theta$), hidden representations, and gradients. Throughout the paper, we use uppercase letters for random variables and lowercase letters for their realizations. For instance, a time series $\mathbf{x}$ and its label $y$ are realizations of the random variables $\mathbf{X} = (X_1, \ldots, X_T)$ and $Y$, respectively. While our primary focus is on explaining black-box predictions, our method can be readily applied for data-centric analysis using the ground-truth label as the explanation target, thereby uncovering the inherent temporal patterns most predictive of the true label.

**Time-series Segments.** Unlike point-wise explanations that assign importance scores to individual time points (*i.e.*, $x_t$), our objective is to find segment-wise explanations. More specifically, we aim to identify a set of contiguous subsequences $\mathbf{x}_{t_1:t_2} \stackrel{\text{def}}{=} (x_{t_1}, \ldots, x_{t_2})$ with $1 \le t_1 \le t_2 \le T$, that are collectively most predictive of the target. To formally denote this, we introduce a *segment-index variable* $s = (t^s, t^e)$, where $t^s, t^e \in [T]$ with $t^s \le t^e$ denote the start and end time points, respectively. The random variable for the corresponding *segment* is then defined as $\mathbf{X}_s = \mathbf{X}_{t^s:t^e}$, with its realization denoted by $\mathbf{x}_s = \mathbf{x}_{t^s:t^e}$. Since explanations often rely on multiple informative regions within a time series, we further extend this notation to an arbitrary number of non-overlapping $K$ segments. Specifically, let $\mathbf{s}_{1:K} = (s_1, \ldots, s_K)$ be an ordered collection of segment-index variables. Then, the corresponding collection of segments can be represented as $\mathbf{X}_{\mathbf{s}_{1:K}} = (\mathbf{X}_{s_1}, \ldots, \mathbf{X}_{s_K})$. We note that the number of segments can vary depending on the input time series.

**Segment-wise Explainer.** For a given instance $\mathbf{X}$, a segment-wise explanation of the black-box model is defined as the subset of segment-index variables that jointly capture the most predictive information about the target black-box outcome $g_\theta(\mathbf{X})$. To achieve this, we introduce a segment-wise explainer, $\mathcal{E}$, which maps an input time series to a set of segment-index variables representing

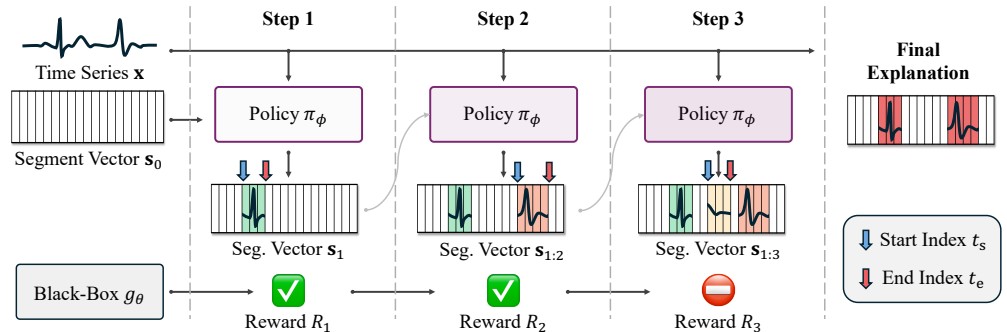

**Figure 2: Overview of TimeSeg.** Given an input time series $\mathbf{x}$, our policy $\pi_\phi$ sequentially samples variable-length segments. At each step $k$, TimeSeg masks $\mathbf{x}$ to obtain $\mathbf{x}_{\mathbf{s}_{1:k}}$ and computes the reward $R_k$ as the *cross-entropy gap*. The procedure terminates when the marginal gain falls below a threshold (or $K_{\max}$ is reached).

candidate combinations of $K$ non-overlapping segments. An *optimal* segment-wise explainer is then obtained by solving the following optimization problem:

**Definition 3.1** (Optimal Segment-wise Explainer). *Given* $\mathbf{X}$*, the optimal segment-wise explainer for* $g_\theta(\mathbf{X})$*, is defined as the solution to:*

$$\mathcal{E}^* = \arg\max_{\mathcal{E}} \; I\big(g_\theta(\mathbf{X}); \mathbf{X}_{\mathbf{s}_{1:K}}\big) - \lambda J\big(\mathbf{s}_{1:K}\big) \quad subject\ to \quad \mathbf{s}_{1:K} \sim \mathcal{E}(\mathbf{X}), \tag{1}$$

*where* $I\big(g_\theta(\mathbf{X}); \mathbf{X}_{\mathbf{s}_{1:K}}\big)$ *denotes the mutual information (MI) that measures the predictive power of the selected segments* $\mathbf{X}_{\mathbf{s}_{1:K}}$ *with respect to* $g_\theta(\mathbf{X})$*, and* $J(\cdot)$ *regularizes the segmentation complexity (e.g., each segment length). The coefficient* $\lambda \in \mathbb{R}^+$ *controls the trade-off between informativity and simplicity of the generated explanations.*

The resulting optimal indices $\mathbf{s}_{1:K}^* = \mathcal{E}^*(\mathbf{x})$ provide an intuitive explanation by highlighting a sparse set of the most salient temporal patterns in $\mathbf{x}$ that the black-box model relies on for predicting $g_\theta(\mathbf{x})$.

### 3.1 Challenges of Obtaining Segment-wise Explanations in Time Series

Solving Eq. (1) would provide an ideal explanation, consisting of *a minimal set of segments* that is *maximally informative* about the target outcome. However, directly optimizing this objective is intractable due to two fundamental challenges: First, accurately estimating the joint mutual information between arbitrary collections of time-series segments and the target outcome is infeasible. Second, identifying the optimal segment-wise explainer requires solving a combinatorial problem over all possible discrete mappings, making an exhaustive search computationally prohibitive.[1]

To jointly address these challenges, we first adopt an *amortized* approach in which segment-index variables are treated as random variables sampled from a distribution parameterized by the explainer for each input time series. This stochastic formulation bypasses the need for direct combinatorial optimization over discrete mappings. We then reformulate the objective by decomposing the joint MI in Eq. (1) into conditional MI (CMI) via the chain rule, and estimate each CMI term as a cross-entropy difference using a variational approximation based on the stochastic explainer. This transformation converts the all-at-once selection of segments into a "sequential decision process". The resulting formulation is not only computationally feasible – replacing exponential search with a tractable procedure – but also enables modeling segments as coherent explanatory units that generalize across different time-series instances. The details of our method are presented in the following section.

## 4 Method: TimeSeg

In this section, we propose a novel information-theoretic segment-wise explainer, which we refer to as TimeSeg. We present the key components of our method in the following order: (Sec. 4.1) reformulating the intractable joint MI optimization as a sequential decision process and casting it

---

[1]Since selecting $K$ segments yields $\binom{T+1}{2K}$ segment candidate sets, summing over all feasible $K$ gives $\sum_{K=1}^{\lfloor (T+1)/2 \rfloor} \binom{T+1}{2K} = 2^T - 1$, which grows exponentially with $T$ (*i.e.*, $\mathcal{O}(2^T)$; see details in Appendix A.3).

within a reinforcement learning (RL) framework, (Sec. 4.2) designing a two-step sampling process that factorizes segment selection into conditional distributions to ensure valid segment generation, (Sec. 4.3) handling the policy gradient optimization with PPO and adaptively determining the optimal number of segments for each instance, and (Sec. 4.4) describing how TimeSeg produces segment explanations end-to-end, from pre-trained black-box predictions to final segment selection.

## 4.1 REFORMULATING SEGMENT SELECTION AS A SEQUENTIAL PROCESS

To address the computational intractability of finding all segments simultaneously, we reformulate the joint MI in Eq. (1) as a sequential decision process that identifies important segments iteratively. Using the chain rule, we decompose the joint MI into a sum of CMI terms as follows:

$$I\big(g_\theta(\mathbf{X}); \mathbf{X}_{\mathbf{s}_{1:K}}\big) = \sum_{k=1}^{K} I\big(g_\theta(\mathbf{X}); \mathbf{X}_{s_k} \mid \mathbf{X}_{\mathbf{s}_{1:k-1}}\big), \tag{2}$$

where the CMI term, $I\big(g_\theta(\mathbf{X}); \mathbf{X}_{s_k} \mid \mathbf{X}_{\mathbf{s}_{1:k-1}}\big)$, measures the information gain about the model's prediction $g_\theta(\mathbf{X})$ from observing the $k$-th segment $\mathbf{X}_{s_k}$, conditioned on the set of previously identified segments $\mathbf{X}_{\mathbf{s}_{1:k-1}}$.

Since the CMI terms in Eq. (2) are intractable to compute directly, we instead maximize a variational approximation derived from the information bottleneck principle (Alemi et al., 2016); see Appendix A.1 for details. The $k$-th conditional mutual information term is given by:

$$I_{\theta,\phi}\big(Y; \mathbf{X}_{s_k} \mid \mathbf{X}_{\mathbf{s}_{1:k-1}}\big) = \mathbb{E}_{\mathbf{x}} \mathbb{E}_{\mathbf{s}_{1:k} \sim \pi_\phi(\cdot \mid \mathbf{x})} \mathbb{E}_{p_\theta(y \mid \mathbf{x})} \Big[\log p_\theta\big(y \mid \mathbf{x}_{\mathbf{s}_{1:k}}\big) - \log p_\theta\big(y \mid \mathbf{x}_{\mathbf{s}_{1:k-1}}\big)\Big]. \tag{3}$$

Here, $p_\theta(y \mid \mathbf{x}_{\mathbf{s}_{1:k}})$ denotes the predictive distribution for class $y$ obtained by the black-box model when its input is restricted to the segments indexed by $\mathbf{s}_{1:k}$, *i.e.*, $g_\theta(\mathbf{x}_{\mathbf{s}_{1:k}})$. The explainer, $\mathcal{E}$, is implemented as a stochastic policy (parameterized by $\phi$), $\pi_\phi$, which sequentially generates segment indices $\mathbf{s}_{1:K}$. At each step $k$, the policy outputs a distribution over the next segment index, conditioned on input $\mathbf{x}$ and previously selected indices $\mathbf{s}_{1:k-1}$, from which $s_k$ is sampled: $s_k \sim \pi_\phi(\cdot \mid \mathbf{x}, \mathbf{s}_{1:k-1})$.

The objective in Eq. (3) can be optimized within a reinforcement learning (RL) framework. We view the explainer as an agent that sequentially interacts with the black-box model. Formally, the state at step $k$ consists of the input $\mathbf{x}$ and the history of previously chosen segment indices $\mathbf{s}_{1:k-1}$. The agent's action is to select the next segment-index variable $s_k \sim \pi_\phi(\cdot \mid \mathbf{x}, \mathbf{s}_{1:k-1})$. The reward for this action is the CMI, which we compute as the cross-entropy gap between the black-box model's outputs with and without augmenting the new segment. This reward is formulated as follows:

$$r_\theta(\mathbf{x}_{s_k}, \mathbf{x}_{\mathbf{s}_{1:k-1}}) \overset{\text{def}}{=} \mathbb{E}_{p_\theta(y \mid \mathbf{x})} \Big[\log p_\theta\big(y \mid \mathbf{x}_{\mathbf{s}_{1:k}}\big) - \log p_\theta\big(y \mid \mathbf{x}_{\mathbf{s}_{1:k-1}}\big)\Big]. \tag{4}$$

To discourage a trivial solution where the policy simply selects all available segments, we introduce a sparsity-inducing cost, $c(s_k)$, into the reward function in Eq. (4). The overall objective is thus to maximize the expected cumulative reward. This aligns our problem with a standard policy optimization framework (Sutton et al., 1998), and we reformulate the final objective as:

$$\mathcal{L}(\phi) = \mathbb{E}_{\mathbf{x}} \mathbb{E}_{\mathbf{s}_{1:K} \sim \pi_\phi(\cdot \mid \mathbf{x})} \left[ \sum_{k=1}^{K} r_\theta\big(\mathbf{x}_{s_k}, \mathbf{x}_{\mathbf{s}_{1:k-1}}\big) - \lambda c\big(s_k\big) \right], \tag{5}$$

where $\lambda$ is the trade-off coefficient introduced in Eq. (1).

## 4.2 STOCHASTIC POLICY FOR SEGMENT-INDEX VARIABLE SELECTION

A key challenge for the policy $\pi_\phi$ is to generate a valid segment-index variable $s_k = (t_k^{\text{s}}, t_k^{\text{e}})$ that satisfies the ordering constraint $t_k^{\text{s}} \leq t_k^{\text{e}}$. To enforce this structurally, we avoid modeling the joint distribution over all valid "start-end" index pairs and instead factor the policy into a product of two conditional distributions: a start-policy $\pi_{\phi^{\text{s}}}$ and an end-policy $\pi_{\phi^{\text{e}}}$. This decomposition is given as

$$\pi_\phi(\mathbf{s} \mid \mathbf{x}, \mathbf{s}_{1:k-1}) \overset{\text{def}}{=} \pi_{\phi^{\text{s}}}(t^{\text{s}} \mid \mathbf{x}, \mathbf{s}_{1:k-1}) \, \pi_{\phi^{\text{e}}}(t^{\text{e}} \mid t^{\text{s}}, \mathbf{x}, \mathbf{s}_{1:k-1}). \tag{6}$$

In this two-step sampling process, our start-policy first samples a *where-to-start* index given the input $\mathbf{x}$ and previously selected segment indices $\mathbf{s}_{1:k-1}$, *i.e.*, $t^{\text{s}} \sim \pi_{\phi^{\text{s}}}(\cdot \mid \mathbf{x}, \mathbf{s}_{1:k-1})$. Then, further

conditioning on this choice, our end-policy selects an *where-to-end* index from the valid range, *i.e.*, $t_k^e \sim \pi_{\phi^e}(\cdot \mid t_k^s, \mathbf{x}, \mathbf{s}_{1:k-1})$. This factorization guarantees by construction that all generated segments are valid and non-empty. In practice, the initial segment $\mathbf{s}_{1:0} = s_0$, where $k = 1$, is initialized as an empty set to provide an initial starting point.

This factorization decomposes the policy $\pi_\phi$ into two *categorical* distributions: $\pi_{\phi^s}$ and $\pi_{\phi^e}$, both defined over absolute time indices. Optimizing this policy requires computing the gradients of the expected reward with respect to $\phi^s$ and $\phi^e$. While individual categorical distributions typically allow continuous relaxations through techniques like Gumbel-Softmax reparameterization – as applied in point-wise explanations (Crabbé & Van Der Schaar, 2021) – the conditional constraint $t_k^s \leq t_k^e$ prevents straightforward application of such reparameterization tricks, motivating the need for a policy gradient method described in the following section.

## 4.3 Learning Segment Selection via Policy Gradient

The primary optimization challenge is that the objective in Eq. (5) is not differentiable with respect to the policy parameters $\phi$, since segment indices are sampled discretely as $s_k \sim \pi_\phi(\cdot \mid \mathbf{x}, \mathbf{s}_{1:k-1})$. While the REINFORCE method (Williams, 1992) gives an unbiased gradient estimate, its direct use is known to suffer from high variance, which often leads to unstable training (Sutton & Barto, 2018).

To improve stability, we adopt an actor–critic framework (Schulman et al., 2015). Specifically, we introduce a *value network* (critic), parameterized by $\psi$, that serves as a learnable baseline. This enables us to compute an *advantage*, $A_k$, which measures the relative benefit of selecting a new segment compared to the baseline, thereby reducing gradient variance. We define the advantage using the one-step temporal difference error: $A_k = R_k + \gamma V_\psi(\mathbf{x}, \mathbf{s}_{1:k}) - V_\psi(\mathbf{x}, \mathbf{s}_{1:k-1})$ where $R_k = r_\theta(\mathbf{x}_{s_k}, \mathbf{x}_{\mathbf{s}_{1:k-1}}) - \lambda c(s_k)$ is the immediate reward of selecting the $k$-th segment and $\gamma \in [0, 1]$ is a discount factor. The value network $V_\psi$ is trained with a mean squared error loss to match a bootstrapped value target. Concretely, we minimize

$$\mathcal{L}_{\text{value}}(\psi) = \mathbb{E}_\mathbf{x}\mathbb{E}_{\mathbf{s}_{1:K} \sim \pi_\phi^{\text{OLD}}}\Big[\big(V_\psi(\mathbf{x}, \mathbf{s}_{1:k-1}) - (R_k + \gamma V_\psi(\mathbf{x}, \mathbf{s}_{1:k}))\big)^2\Big], \tag{7}$$

such that $V_\psi$ approximates the expected return and the resulting advantage $A_k$ provides a low-variance baseline for the policy-gradient update. Furthermore, to regulate the magnitude of policy updates, we optimize the policy $\pi_\phi$ (actor) using the clipped surrogate objective of *proximal policy optimization* (PPO) (Schulman et al., 2017). Then, the final objective can be given as follows:

$$\mathcal{L}_{\text{PPO}}(\phi) = \mathbb{E}_\mathbf{x}\mathbb{E}_{\mathbf{s}_{1:K} \sim \pi_\phi^{\text{OLD}}}\left[\sum_{k=1}^{K} \min\Big(\rho_k \cdot A_k, \text{CLIP}(\rho_k, 1 - \epsilon, 1 + \epsilon)A_k\Big)\right], \tag{8}$$

where $\rho_k = \frac{\pi_\phi(s_k|\mathbf{x}, \mathbf{s}_{1:k-1})}{\pi_\phi^{\text{OLD}}(s_k|\mathbf{x}, \mathbf{s}_{1:k-1})}$ is the sampling ratio.

The policy and value networks are trained jointly, providing a robust and stable optimization procedure. Additional details are provided in Appendix A.2.

**Determining the Number of Selected Segments.** Since the optimal number of segments can vary across time series, we employ an instance-specific termination criterion to decide when to stop the selection process. At each step $k$, we compute the CMI reward in Eq. (4), which is the marginal information gain (*i.e.*, the difference in cross-entropy), from adding the newly proposed segment. We normalize it with the current cross-entropy and terminate the selection procedure when this relative reward falls below a predefined threshold $\tau$:

$$r_\theta(\mathbf{x}_{s_k}, \mathbf{x}_{\mathbf{s}_{1:k-1}})/\mathbb{E}_{p_\theta(y|\mathbf{x})}\big[-\log p_\theta(y \mid \mathbf{x}_{\mathbf{s}_{1:k-1}})\big] \leq \tau. \tag{9}$$

This allows TimeSeg to adaptively determine the number of selected segments $K$ and produce instance-specific explanations. In our experiments, we set $\tau = 0.3$ and $K_{\max} = 5$.

## 4.4 Segment Representation via Gating Vectors

At each selection step, the policy network $\pi_\phi$ and value network $V_\psi$ are provided with the entire sequence $\mathbf{x}$ together with the previously selected segment indices $\mathbf{s}_{1:k-1}$. This full-context access allows these networks to assess the global structure of the time series and strategically search for the

**Table 1:** Performance comparison on four datasets with ground-truth; **bold** indicates best, underlined indicates second-best, and the asterisk (*) denotes methods that violate the strict black-box setup.

| Method | SeqComb-UV | | | LowVarDetect-UV | | |
|---|---|---|---|---|---|---|
| | F1 ↑ | IoU ↑ | Cont. ↓ | F1 ↑ | IoU ↑ | Cont. ↓ |
| IG* | 0.539±0.059 | 0.379±0.057 | 0.166±0.022 | **0.582±0.089** | **0.421±0.089** | 0.170±0.065 |
| Dynamask | 0.157±0.016 | 0.088±0.010 | 0.215±0.036 | 0.177±0.046 | 0.115±0.038 | 0.181±0.074 |
| WinIT | 0.208±0.033 | 0.123±0.021 | 0.178±0.031 | 0.383±0.022 | 0.242±0.017 | 0.117±0.024 |
| LIMESegment | 0.430±0.009 | 0.289±0.008 | **0.009±0.001** | 0.467±0.063 | 0.314±0.058 | **0.012±0.001** |
| TimeX++* | 0.636±0.157 | 0.489±0.159 | 0.098±0.015 | 0.432±0.082 | 0.291±0.057 | 0.231±0.025 |
| **TimeSeg** | **0.645±0.018** | **0.495±0.018** | 0.017±0.000 | 0.499±0.023 | 0.356±0.015 | 0.020±0.004 |

| Method | FreqShapes-V | | | MIT-ECG | | |
|---|---|---|---|---|---|---|
| | F1 ↑ | IoU ↑ | Cont. ↓ | F1 ↑ | IoU ↑ | Cont. ↓ |
| IG* | 0.687±0.003 | 0.534±0.004 | 0.101±0.003 | 0.589±0.006 | 0.435±0.007 | 0.056±0.007 |
| Dynamask | 0.103±0.007 | 0.060±0.004 | 0.144±0.001 | 0.353±0.023 | 0.221±0.018 | 0.072±0.016 |
| WinIT | 0.454±0.005 | 0.307±0.005 | 0.114±0.001 | 0.241±0.062 | 0.147±0.042 | 0.133±0.014 |
| LIMESegment | 0.440±0.004 | 0.287±0.003 | **0.030±0.001** | 0.491±0.068 | 0.359±0.063 | 0.006±0.001 |
| TimeX++* | **0.799±0.001** | **0.666±0.002** | 0.120±0.001 | 0.593±0.146 | 0.460±0.142 | 0.016±0.002 |
| **TimeSeg** | 0.722±0.014 | 0.576±0.018 | 0.085±0.001 | **0.739±0.016** | **0.621±0.021** | **0.006±0.000** |

next informative segment. In contrast, the black-box time-series classifier $g_\theta$ only observes a set of segments $\mathbf{x}_{\mathbf{s}_{1:k}}$, matching the limited view that will ultimately be provided as the explanation.

To encode this asymmetric access, we introduce *binary gate vectors*, $\mathbf{m}_1, \ldots, \mathbf{m}_K$, where each $\mathbf{m}_k = (m_{k,1}, \ldots, m_{k,T}) \in \{0,1\}^T$ indicates whether a time point $t$ is covered by the $k$-th selected segments: $m_{k,t} = 1$ if $t \in \{t_k^s, \ldots, t_k^e\}$ and $m_{k,t} = 0$ otherwise. For the first $k$ selections, we define the *combined* gate vector $\mathbf{m}_{1:k} = \mathbf{m}_1 \vee \mathbf{m}_2 \vee \cdots \vee \mathbf{m}_k$, where $\vee$ is an element-wise OR operator, which indicates whether each time point $t$ is covered by any of the first $k$ selected segments.

For the policy and value networks, we concatenate $\mathbf{m}_{1:k-1}$ to $\mathbf{x}$ along the feature axis, forming an augmented input $[\mathbf{x}; \mathbf{m}_{1:k-1}]$. This design provides full temporal context while explicitly marking which segments have already been selected. Meanwhile, to ensure that the black-box model makes predictions only based on the selected segments, we apply the gate vector to the original sequence as $\mathbf{x}_{\mathbf{s}_{1:k}} = \mathbf{m}_{1:k} \odot \mathbf{x} + (1 - \mathbf{m}_{1:k}) \odot \bar{\mathbf{x}}$, where $\odot$ is element-wise multiplication and $\bar{\mathbf{x}}$ is the empirical mean of the training set, used to neutralize unselected time points.

**Sparsity-inducing Regularization.** Using the gate vectors, we can formulate the sparsity regularization introduced in Eq. (1), which encourages the selection of compact and non-redundant segments. Specifically, at the $k$-th selection step, the sparsity cost, $c(s_k) = \frac{1}{T}\|\mathbf{m}_k\|_1$, is given by the length of the chosen segments, which is then integrated into the immediate reward as a penalty term: $R_k = r_\theta(\mathbf{x}_{s_k}, \mathbf{x}_{\mathbf{s}_{1:k-1}}) - \lambda c(s_k)$.

**Segment Selection.** At training time, we sample the start and end indices from the policy's output distribution to enable exploration and facilitate optimization. In contrast, at inference time, we use only the policy network and select segments deterministically via argmax. This produces stable and deterministic explanations. The process repeats until the termination criterion is satisfied.

## 5 EXPERIMENTS

**Datasets.** We evaluate TimeSeg on both synthetic and real-world time series datasets across three categories: (i) synthetic datasets with ground-truth explanatory segments, (ii) real-world datasets with segment-level annotations as ground-truth explanations, and (iii) real-world datasets without ground truth explanations. For the synthetic datasets, we adapt datasets from TimeX (Queen et al., 2023): **SeqComb-UV** (used as originally released), **FreqShapes-V** (a variant with duplicate class-defining segments removed to ensure single explanatory patterns per sequence), and **LowVarDetect-UV** (univariate adaptation of the multivariate LowVar dataset). For annotated real-world data, we use the **MIT-ECG** (Moody & Mark, 2001) arrhythmia detection dataset. For unannotated evaluation, we include the **Epilepsy** EEG seizure detection (Andrzejak et al., 2002), **Wafer** (Olszewski, 2001) and **GunPoint** (Ratanamahatana & Keogh, 2005). Additional details are provided in Appendix C.

**Table 2:** Occlusion analysis on four datasets under two substitution strategies (Mean / Zero). For each baseline and dataset, we report AUROC drop ratio when retaining only selected segments (Suff. ↓) and when removing them (Comp. ↑), along with a Contiguity score (Cont. ↓).

| Method | MIT-ECG | | | | | Epilepsy | | | | |
|---|---|---|---|---|---|---|---|---|---|---|
| | Mean | | Zero | | Cont.↓ | Mean | | Zero | | Cont.↓ |
| | Suff.↓ | Comp.↑ | Suff.↓ | Comp.↑ | | Suff.↓ | Comp.↑ | Suff.↓ | Comp.↑ | |
| Random | $55.21_{\pm3.45}$ | $0.01_{\pm0.01}$ | $53.63_{\pm2.20}$ | $0.02_{\pm0.01}$ | $0.07_{\pm0.01}$ | $3.12_{\pm1.37}$ | $0.04_{\pm0.07}$ | $\underline{4.17}_{\pm1.82}$ | $0.04_{\pm0.05}$ | $0.22_{\pm0.02}$ |
| WinIT | $\underline{23.51}_{\pm9.01}$ | $0.49_{\pm0.30}$ | $\underline{23.51}_{\pm9.01}$ | $0.49_{\pm0.30}$ | $0.09_{\pm0.01}$ | $\underline{1.99}_{\pm0.31}$ | $0.09_{\pm0.08}$ | $\mathbf{2.63}_{\pm0.42}$ | $0.11_{\pm0.08}$ | $0.20_{\pm0.02}$ |
| Dynamask | $45.31_{\pm7.47}$ | $2.51_{\pm1.56}$ | $46.05_{\pm6.64}$ | $2.61_{\pm1.51}$ | $0.02_{\pm0.00}$ | $3.06_{\pm1.29}$ | $\underline{0.60}_{\pm0.79}$ | $4.24_{\pm0.93}$ | $\underline{0.62}_{\pm0.85}$ | $0.14_{\pm0.01}$ |
| TimeX++* | $40.31_{\pm7.52}$ | $\underline{5.31}_{\pm3.54}$ | $42.52_{\pm6.05}$ | $\mathbf{5.79}_{\pm3.67}$ | $\mathbf{0.01}_{\pm0.00}$ | $6.79_{\pm9.46}$ | $0.07_{\pm0.11}$ | $13.09_{\pm20.79}$ | $0.09_{\pm0.10}$ | $\underline{0.07}_{\pm0.02}$ |
| **TimeSeg** | $\mathbf{0.70}_{\pm0.23}$ | $\mathbf{5.54}_{\pm1.96}$ | $\mathbf{1.94}_{\pm0.40}$ | $\underline{4.92}_{\pm1.38}$ | $\mathbf{0.01}_{\pm0.00}$ | $\mathbf{1.94}_{\pm0.60}$ | $\mathbf{0.93}_{\pm0.85}$ | $7.49_{\pm2.82}$ | $\mathbf{0.73}_{\pm0.42}$ | $\mathbf{0.01}_{\pm0.00}$ |

| Method | Wafer | | | | | GunPoint | | | | |
|---|---|---|---|---|---|---|---|---|---|---|
| | Mean | | Zero | | Cont.↓ | Mean | | Zero | | Cont.↓ |
| | Suff.↓ | Comp.↑ | Suff.↓ | Comp.↑ | | Suff.↓ | Comp.↑ | Suff.↓ | Comp.↑ | |
| Random | $34.23_{\pm4.53}$ | $0.01_{\pm0.00}$ | $34.23_{\pm4.53}$ | $0.01_{\pm0.00}$ | $0.18_{\pm0.02}$ | $32.42_{\pm4.47}$ | $1.42_{\pm0.19}$ | $32.42_{\pm4.47}$ | $1.42_{\pm0.19}$ | $0.31_{\pm0.03}$ |
| WinIT | $49.12_{\pm12.52}$ | $\underline{1.95}_{\pm0.43}$ | $49.12_{\pm12.52}$ | $\underline{1.95}_{\pm0.43}$ | $0.05_{\pm0.00}$ | $\underline{32.39}_{\pm10.48}$ | $1.88_{\pm1.07}$ | $\underline{32.39}_{\pm10.48}$ | $1.88_{\pm1.07}$ | $0.14_{\pm0.03}$ |
| Dynamask | $31.13_{\pm3.59}$ | $1.59_{\pm1.10}$ | $31.13_{\pm3.59}$ | $1.59_{\pm1.10}$ | $0.05_{\pm0.00}$ | $35.02_{\pm1.14}$ | $\underline{9.15}_{\pm4.81}$ | $35.02_{\pm1.14}$ | $\underline{9.15}_{\pm4.81}$ | $0.10_{\pm0.01}$ |
| TimeX++* | $52.39_{\pm8.13}$ | $0.22_{\pm0.20}$ | $52.39_{\pm8.13}$ | $0.22_{\pm0.20}$ | $\underline{0.03}_{\pm0.02}$ | $45.41_{\pm8.95}$ | $7.10_{\pm6.45}$ | $45.41_{\pm8.95}$ | $7.10_{\pm6.45}$ | $\mathbf{0.01}_{\pm0.00}$ |
| **TimeSeg** | $\mathbf{0.19}_{\pm0.07}$ | $\mathbf{2.42}_{\pm1.09}$ | $\mathbf{0.19}_{\pm0.07}$ | $\mathbf{2.42}_{\pm1.09}$ | $\mathbf{0.01}_{\pm0.01}$ | $\mathbf{1.44}_{\pm1.27}$ | $\mathbf{46.41}_{\pm10.44}$ | $\mathbf{1.43}_{\pm1.28}$ | $\mathbf{46.41}_{\pm10.44}$ | $\underline{0.02}_{\pm0.00}$ |

**Benchmark Methods.** We compare TimeSeg against a set of state-of-the-art time series explainers, including gradient-based methods such as **Integrated Gradients (IG)** (Sundararajan et al., 2017), point-wise attribution methods including **Dynamask** (Crabbé & Van Der Schaar, 2021), **WinIT** (Leung et al.), and **TimeX++** (Liu et al., 2024), as well as the patch-wise method **LIMESegment** (Sivill & Flach, 2022). While IG requires direct access to model gradients and TimeX++ depends on internal embeddings and architectural knowledge, TimeSeg operates in a *strict black-box setting* using only model inputs and outputs (requiring neither gradients nor internal representations). These methods are therefore *evaluated under their optimal conditions*, ensuring a rigorous and fair comparison.

**Evaluation.** We evaluate the quality of explanations along three complementary dimensions, with all results reported as mean ± standard deviation over 5-fold cross-validation:

- **Overlap with Ground Truth**: For datasets with ground-truth segments (*i.e.*, synthetic benchmarks and MIT-ECG), we measure the overlap between the selected temporal regions and the ground truth. We report the F1-score and Jaccard index (IoU) to quantify this overlap.

- **Explanation Fidelity**: To assess whether explanations correctly identify temporal regions used by the black-box model, we use two complementary metrics: (i) Sufficiency (↓) measures the drop in AUROC when using only the selected segments while masking the remainder, and (ii) Comprehensiveness (↑) measures AUROC drop when removing the selected segments from the original sequence. Masked regions are replaced with zero or the empirical mean.

- **Segment Quality**: We assess the quality of segment-based explanations using (i) Sparsity (↓), the fraction of selected regions relative to total sequence length (lower values indicate more concise explanations) and (ii) Contiguity (↓), which counts transitions between masked and unmasked regions, normalized by sequence length. Lower values indicate fewer segment boundaries, leading to less complex and more concise explanations.

**Implementation Details.** The target black-box model, $g_\theta$, is implemented as a Temporal Convolutional Network (TCN) (Lea et al., 2017). For our method, both the policy, $\pi_\phi$, and the value network, $V_\psi$, are implemented as 3-layer 1D CNNs. We provide the full details in Appendix C.

## 5.1 QUANTITATIVE ANALYSIS

**Evaluation with Ground-Truth Annotations.** We evaluate TimeSeg on datasets with segment-level ground truth annotations, including three synthetic datasets (SeqComb-UV, LowVarDetect-UV, FreqShapes-V) and one real-world dataset (MIT-ECG). Here, all evaluated methods are configured to achieve an equal level of sparsity within each dataset (*i.e.*, SeqComb-UV: 0.14, FreqShape-V: 0.10, LowVarDetect-UV: 0.18, and MIT-ECG: 0.14). As shown in Table 1, TimeSeg consistently achieves the best or second-best performance across all metrics. In particular, TimeSeg substantially outperforms other methods on the MIT-ECG dataset, achieving F1 and IOU scores of 0.739 and 0.621 (compared to the second-best benchmark, TimeX++, which achieves 0.593 and 0.460, respec-

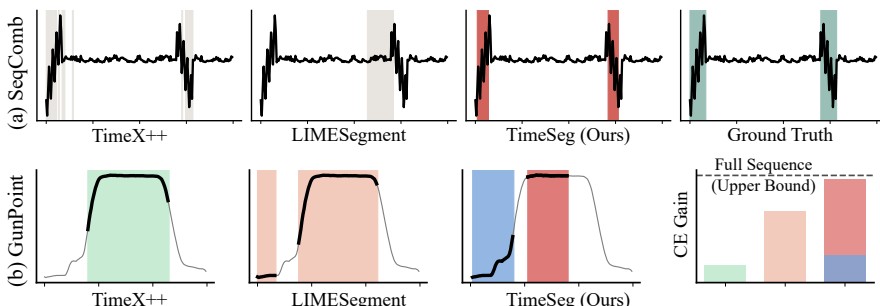

Figure 3: Qualitative Analysis on the SeqComb and GunPoint Datasets.

tively). This improvement is especially remarkable since TimeSeg operates in a strict black-box setting, relying solely on model inputs and outputs, whereas TimeX++ and IG utilize internal gradients and embeddings of the target black-box model. Regarding segment quality, TimeSeg significantly outperforms point-wise methods and achieves contiguity on par with LIMESegment. However, unlike LIMESegment, which struggles with pinpointing important temporal regions (evidenced by low F1 and IOU), TimeSeg produces segments that closely align with ground-truth annotations.

**Evaluation without Ground-Truth Annotations.** Since real-world datasets lack segment-level annotations, we measure the performance drop (*i.e.*, AUROC) of the target black-box model using Sufficiency and Comprehensiveness. Here, all evaluated methods are configured to achieve an equal level of sparsity within each dataset (*i.e.*, MIT-ECG: 0.14, Epilepsy: 0.10, Wafer: 0.09, and Gun-Point: 0.19). As shown in Table 2, TimeSeg achieves the best performance across nearly all datasets and metrics, with particularly strong gains on the Wafer and GunPoint datasets. More specifically, TimeSeg shows $\leq 2\%$ drop when the target black-box model only utilizes the selected segments for prediction, whereas the second-best method suffers a drop of $\geq 31\%$. Moreover, these results are obtained with highly coherent segment-wise explanations, achieving contiguity scores as low as $1 - 2\%$, indicating the selected segments are both concise and well-connected.

## 5.2 QUALITATIVE ANALYSIS

Figure 3 presents a qualitative analysis of segments identified on the synthetic Seq-Comb dataset and the real-world GunPoint dataset. The classification task in the Se-qComb dataset poses a unique challenge, requiring the explainer to accurately re-cover two distinct segments critical to the class label. Our analysis reveals notable differences among methods: (i) TimeX++ correctly pinpoints relevant regions but generates fragmented explanations, lack-

**Table 3:** Ablation study on $\lambda$ for the MIT-ECG dataset. We report F1, IoU, Sufficiency, and average Sparsity.

| $\lambda$ | F1 ↑ | IoU ↑ | Suff. ↓ | Sparsity ↓ |
|---|---|---|---|---|
| 0.1 | 0.702±0.042 | 0.577±0.054 | **0.558±0.023** | 0.218±0.023 |
| 0.3 | **0.739±0.016** | **0.621±0.021** | 0.704±0.232 | 0.144±0.018 |
| 0.5 | 0.727±0.020 | 0.608±0.027 | 0.757±0.014 | 0.123±0.137 |
| 0.7 | 0.721±0.053 | 0.609±0.053 | 2.442±3.113 | 0.110±0.012 |
| 0.9 | 0.690±0.035 | 0.592±0.039 | 4.728±2.340 | **0.098±0.007** |

ing coherence. (ii) LIMESegment produces contiguous segments; however, its reliance on fixed-size patches often results in misaligned explanations. In contrast, TimeSeg consistently identifies complete, well-aligned segments that closely match the ground-truth annotations.

For the GunPoint dataset, which lacks ground-truth annotations, we compare the selected temporal regions by measuring the cross-entropy gain of the black-box classifier when using the selected segments as an input. The dashed line in Figure 3 indicates the cross-entropy achieved with the full time series, serving as an upper bound. Although TimeSeg and TimeX++ select segments of comparable total length, TimeSeg achieves a cross-entropy gain near this upper bound, while TimeX++ exhibits substantially lower gains. LIMESegment selects segments that are roughly 25% longer than those of TimeSeg but still shows a lower cross-entropy gain, underscoring our method's superior segment localization. Additional examples in Appendix D.3 consistently demonstrate similar trends.

**Table 4:** Ablation study on $K_{\max}$ for the SeqComb-UV dataset. We report explanation metrics (Sparsity, F1, IoU, Contiguity) and structural statistics, including the average number of segments and segment length.

| $K_{\max}$ | Sparsity | F1 | IoU | Contiguity | # Segments | Segment Length |
|---|---|---|---|---|---|---|
| 1 | $0.188 \pm 0.017$ | $0.499 \pm 0.013$ | $0.343 \pm 0.012$ | $0.010 \pm 0.000$ | $1.000 \pm 0.000$ | $37.549 \pm 3.473$ |
| 2 | $0.170 \pm 0.022$ | $0.643 \pm 0.024$ | $0.494 \pm 0.028$ | $0.016 \pm 0.001$ | $1.619 \pm 0.056$ | $21.751 \pm 3.416$ |
| 3 | $0.154 \pm 0.017$ | $0.652 \pm 0.011$ | $0.504 \pm 0.010$ | $0.016 \pm 0.001$ | $1.647 \pm 0.060$ | $18.102 \pm 2.412$ |
| 5 | $0.141 \pm 0.016$ | $0.645 \pm 0.018$ | $0.497 \pm 0.018$ | $0.017 \pm 0.000$ | $1.631 \pm 0.060$ | $17.615 \pm 2.869$ |
| 7 | $0.140 \pm 0.025$ | $0.642 \pm 0.015$ | $0.492 \pm 0.017$ | $0.016 \pm 0.001$ | $1.647 \pm 0.061$ | $17.610 \pm 3.789$ |

## 5.3 Ablation Study: The Effect of $\lambda$ and $K_{\max}$

We investigate the impact of the sparsity-inducing coefficient $\lambda$ on explanation quality and predictive performance, as well as the maximum number of segments $K_{\max}$ on the structure of the selected segments. Results obtained with varying $\lambda$ on the MIT-ECG dataset and $K_{\max}$ on the SeqComb-UV dataset are reported in Table 3 and Table 4, respectively. As $\lambda$ increases, the explainer produces more compact segments with a slight drop in predictive performance. Conversely, smaller values of $\lambda$ lead to longer segments that better preserve predictive performance at the cost of reduced conciseness. These results indicate that the sparsity-inducing coefficient behaves as designed, providing intuitive control over the sparsity structure of the generated explanations. Regarding $K_{\max}$, when it is set to a very small value (e.g., $K_{\max} = 1$), the policy is constrained to select a single extended segment. However, once $K_{\max} \geq 3$, the policy no longer reaches the upper bound, and metrics become stable. This suggests that $K_{\max}$ need only be sufficiently large to avoid constraining the policy, rather than finely tuned.

## 5.4 Additional Analysis

We provide additional analysis to examine the robustness, extensibility, and practical behavior of TimeSeg. Appendix D.4 also analyzes sensitivity to the termination threshold $\tau$, showing stable performance over broad ranges with intuitive changes in segment structure. Appendix D.5 extends TimeSeg to multivariate time series via channel selection, achieving comparable performance while maintaining strong contiguity. Appendix D.6 demonstrates consistent performance across RNN and Transformer black-box backbones, confirming architecture-agnostic behavior. Appendix D.7 reports computational cost and training dynamics, highlighting efficient interaction budgets and stable convergence. Lastly, Appendix D.8 presents DTW-based clustering of extracted segments, revealing recurring and semantically meaningful patterns.

## 6 Conclusion

In this paper, we introduced TimeSeg, an information-theoretic framework for segment-wise explanations of time-series black-box models. Our work first defines what a meaningful segment is in time series, and formulates the segment selection problem as a sequential decision process, enabling the identification of concise and informative temporal segments under a strict black-box setting. Through extensive experiments on synthetic and real-world datasets, we demonstrated that TimeSeg produces more interpretable explanations than existing point-wise and patch-wise approaches, while maintaining competitive or superior explanation fidelity.

**Future Work.** Building upon our framework, one promising direction is to incorporate more sophisticated policy optimization techniques, distributional RL objectives, or model-based exploration strategies to further improve stability and efficiency of model training. In addition, the *reward design* can be extended beyond purely information-theoretic criteria to reflect human preferences for explanations or based on domain knowledge — priors that encourage clinically meaningful segments. This opens the door to human-in-the-loop training (*i.e.*, preference learning), multi-objective formulations that trade off fidelity and readability, and application-specific constraints that tailor explanations to domain conventions. We believe that bridging explainable AI for time series with both the broader toolbox of RL methods *and* human-centric reward shaping offers a promising direction for developing more general and useful explanation models.

ACKNOWLEDGEMENTS

We thank the ICLR reviewers for their comments and suggestions. This work is supported by the National Research Foundation of Korea (NRF) grant funded by the Korea government (MSIT) (No. RS-2024-00358602) and the Institute of Information & Communications Technology Planning & Evaluation (IITP) grant funded by the Korea government (MSIT), Artificial Intelligence Graduate School Program (No. RS-2019-II190079, Korea University), the Artificial Intelligence Star Fellowship Support Program to nurture the best talents (No. RS-2025-02304828), and the AI Research Hub Project (No. RS-2024-00457882).

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

# A  MATHEMATICAL DERIVATIONS

## A.1  DERIVATIONS FOR MUTUAL INFORMATION

We derive the objective used in Eq. (1), starting from the joint mutual information between the target $Y$ and the selected segment set $\mathbf{X}_{\mathbf{s}_{1:K}}$ and then obtaining a tractable variational lower bound for each conditional term. In our setting the target is the black-box prediction $g_\theta(\mathbf{X})$, but for brevity we write it as $Y$ in this appendix.

**Chain Rule for Mutual Information.** For random variables $Y$ and a ordered tuple of segment variables $\mathbf{X}_{\mathbf{s}_{1:K}} = (\mathbf{X}_{s_1}, \ldots, \mathbf{X}_{s_K})$, the chain rule gives

$$I(Y; \mathbf{X}_{\mathbf{s}_{1:K}}) = I(Y; \mathbf{X}_{s_1}, \ldots, \mathbf{X}_{s_K}) \tag{10}$$

$$= I(Y; \mathbf{X}_{s_1}) + I(Y; \mathbf{X}_{s_2}, \cdots \mathbf{X}_{s_K} \mid \mathbf{X}_{s_1}) \tag{11}$$

$$= \cdots$$

$$= I(Y; \mathbf{X}_{s_1}) + I(Y; \mathbf{X}_{s_2} \mid \mathbf{X}_{s_1}) \tag{12}$$

$$+ \cdots + I(Y; \mathbf{X}_{s_k} \mid \mathbf{X}_{s_1}, \cdots \mathbf{X}_{s_{k-1}})$$

$$= \sum_{k=1}^{K} I(Y; \mathbf{X}_{s_k} \mid \mathbf{X}_{\mathbf{s}_{1:k-1}}). \tag{13}$$

with $\mathbf{s}_{1:0} = \emptyset$ and $I(Y; \mathbf{X}_{s_1} \mid \mathbf{X}_{\mathbf{s}_{1:0}}) = I(Y; \mathbf{X}_{s_1})$. This is the chain-rule decomposition used in Eq. (2) of the main text when $Y$ is replaced by the black-box prediction $g_\theta(\mathbf{X})$.

**Variational Lower Bound for Conditional Mutual Information.** By definition, the conditional mutual information is

$$I(Y; \mathbf{X}_{s_k} \mid \mathbf{X}_{\mathbf{s}_{1:k-1}}) = \mathbb{E}_{\mathbf{x}_{\mathbf{s}_{1:k}}, y} \left[ \log \frac{p(y, \mathbf{x}_{s_k} \mid \mathbf{x}_{\mathbf{s}_{1:k-1}})}{p(y \mid \mathbf{x}_{\mathbf{s}_{1:k-1}}) \cdot p(\mathbf{x}_{s_k} \mid \mathbf{x}_{\mathbf{s}_{1:k-1}})} \right] \tag{14}$$

where $\mathrm{KL}(\cdot\|\cdot)$ denotes the Kullback–Leibler divergence and the conditionals $p(y \mid \mathbf{x}_{\mathbf{s}_{1:k}})$ and $p(y \mid \mathbf{x}_{\mathbf{s}_{1:k-1}})$ are induced by the black-box predictor $g_\theta$ (we drop the subscript $\theta$ for brevity). However, because the black-box setting precludes direct access to the densities $p(\mathbf{x}_{s_k} \mid \mathbf{x}_{\mathbf{s}_{1:k-1}})$ and $p(y, \mathbf{x}_{s_k} \mid \mathbf{x}_{\mathbf{s}_{1:k-1}})$. we rewrite the objective as below.

$$I(Y; \mathbf{X}_{s_k} \mid \mathbf{X}_{\mathbf{s}_{1:k-1}}) = \mathbb{E}_{\mathbf{x}_{\mathbf{s}_{1:k}}, y} \left[ \log p(y \mid \mathbf{x}_{s_k}, \mathbf{x}_{\mathbf{s}_{1:k-1}}) - \log p(y \mid \mathbf{x}_{\mathbf{s}_{1:k-1}}) \right] \tag{15}$$

Lastly, we approximate the marginal $\mathbf{X}_{\mathbf{s}_{1:k}}$ over $\mathbf{X}$ by sampling segment indices from the policy $\pi_\phi$ and treating the segment as the induced subsequence of $\mathbf{x}$, $\mathbf{s}_{1:k} \sim \pi_\phi(\cdot \mid \mathbf{x})$.

$$I_{\theta,\phi}(Y; \mathbf{X}_{s_k} \mid \mathbf{X}_{\mathbf{s}_{1:k-1}}) = \mathbb{E}_{\mathbf{x}} \mathbb{E}_{\mathbf{s}_{1:k} \sim \pi_\phi(\cdot|\mathbf{x})} \mathbb{E}_{y|\mathbf{x}} \left[ \log p(y \mid \mathbf{x}_{\mathbf{s}_{1:k}}) - \log p(y \mid \mathbf{x}_{\mathbf{s}_{1:k-1}}) \right], \tag{16}$$

which we maximize with respect to $\phi$ (e.g., via policy-gradient methods), while $p(\cdot \mid \cdot)$ remains fixed and is provided by the black-box predictor $g_\theta$.

## A.2  REINFORCEMENT LEARNING OPTIMIZATION DETAILS

**Policy Gradient with Discrete Segment Sampling.** Our training objective in Eq. (5) can be rewritten more simplicity as

$$\mathcal{L}(\phi) = \mathbb{E}_{\mathbf{x}} \mathbb{E}_{\mathbf{s}_{1:K} \sim \pi_\phi(\cdot|\mathbf{x})} \left[ \sum_{k=1}^{K} R_k \right], \tag{17}$$

where $R_k = r_\theta(\mathbf{x}_{s_k}, \mathbf{x}_{\mathbf{s}_{1:k-1}}) - \lambda c(s_k, \mathbf{s}_{1:k-1})$ denotes the conditional mutual information reward, and $c(\cdot)$ the sparsity cost. The policy $\pi_\phi$ generates a trajectory of segment indices $\mathbf{s}_{1:K}$, and the objective is to maximize the expected cumulative reward under this distribution.

Since the actions $s_k$ are discrete start–end indices, gradients cannot propagate directly through the sampling step. Instead, we apply the policy gradient methods to obtain the gradient respect to $\phi$:

$$\nabla_\phi \mathcal{L}(\phi) = \nabla_\phi \mathbb{E}_\mathbf{x} \mathbb{E}_{\mathbf{s}_{1:K} \sim \pi_\phi} \left[ \sum_{k=1}^K R_k \right] \tag{18}$$

$$= \mathbb{E}_\mathbf{x} \left[ \int \left( \sum_{k=1}^K R_k \right) \cdot \nabla_\phi \pi_\phi(\mathbf{s}_{1:K} \mid \mathbf{x}) \, d\mathbf{s}_{1:K} \right] \tag{19}$$

$$= \mathbb{E}_\mathbf{x} \left[ \int \left( \sum_{k=1}^K R_k \right) \cdot \nabla_\phi \pi_\phi(\mathbf{s}_{1:K} \mid \mathbf{x}) \cdot \frac{\pi_\phi(\mathbf{s}_{1:K} \mid \mathbf{x})}{\pi_\phi(\mathbf{s}_{1:K} \mid \mathbf{x})} \, d\mathbf{s}_{1:K} \right] \tag{20}$$

$$= \mathbb{E}_\mathbf{x} \left[ \int \left( \sum_{k=1}^K R_k \right) \cdot \nabla_\phi \log \pi_\phi(\mathbf{s}_{1:K} \mid \mathbf{x}) \cdot \pi_\phi(\mathbf{s}_{1:K} \mid \mathbf{x}) \, d\mathbf{s}_{1:K} \right] \tag{21}$$

$$= \mathbb{E}_\mathbf{x} \mathbb{E}_{\mathbf{s}_{1:K} \sim \pi_\phi} \left[ \nabla_\phi \log \pi_\phi(\mathbf{s}_{1:K} \mid \mathbf{x}) \cdot \sum_{k=1}^K R_k \right] \tag{22}$$

$$= \mathbb{E}_\mathbf{x} \mathbb{E}_{\mathbf{s}_{1:K} \sim \pi_\phi} \left[ \sum_{k=1}^K \left( R_k \cdot \nabla_\phi \log \pi_\phi(s_k \mid \mathbf{x}, \mathbf{s}_{1:k-1}) \right) \right] \tag{23}$$

$$\tag{24}$$

This derivation shows that policy optimization reduces to weighting the log-likelihood gradient of the chosen segment $s_k$ by the observed reward $R_k$. Although unbiased, this estimator typically suffers from high variance, motivating the use of variance reduction techniques such as actor–critic methods and proximal policy optimization (PPO).

**Advantage in the Actor–Critic Framework.** To reduce variance, we employ an actor–critic framework (Schulman et al., 2015). The policy $\pi_\phi$ plays the role of the actor, while we introduce a value network $V_\psi(\mathbf{x}, \mathbf{s}_{1:k})$, parameterized by $\psi$, as the critic. The critic estimates the expected cumulative return given the current sequence of selected segments, e.g.,

$$V_\psi(\mathbf{x}, \mathbf{s}_{1:k-1}) \approx \mathbb{E}_{\pi_\phi} \left[ \sum_{t=k}^K \gamma^{t-k} R_t \;\middle|\; \mathbf{x}, \mathbf{s}_{1:k-1} \right],$$

This allows us to compute the *advantage* function:

$$A_k = R_k + \gamma V_\psi(\mathbf{x}, \mathbf{s}_{1:k}) - V_\psi(\mathbf{x}, \mathbf{s}_{1:k-1}), \tag{25}$$

where $\gamma \in [0, 1]$ is a discount factor. The advantage measures how much better the observed return is compared to the critic's baseline estimate. Replacing raw rewards $R_k$ with advantages $A_k$ significantly reduces gradient variance without introducing bias.

The value network is trained jointly with the policy using a temporal difference (TD) loss:

$$\mathcal{L}_{\text{value}}(\psi) = \mathbb{E}_{\mathbf{x}, \mathbf{s}_{1:k} \sim \pi_\phi} \left[ \left( V_\psi(\mathbf{x}, \mathbf{s}_{1:k-1}) - (R_k + \gamma V_\psi(\mathbf{x}, \mathbf{s}_{1:k})) \right)^2 \right]. \tag{26}$$

**Stabilization via Proximal Policy Optimization.** Although advantage estimates stabilize the gradient, policy updates can still be unstable if the new policy $\pi_\phi$ diverges too far from the old one $\pi_\phi^{\text{old}}$. To address this, we adopt the PPO objective (Schulman et al., 2017), which clips large changes in the policy update:

$$\mathcal{L}_{\text{PPO}}(\phi) = \mathbb{E}_\mathbf{x} \mathbb{E}_{\mathbf{s}_{1:K} \sim \pi_\phi^{\text{OLD}}} \left[ \sum_{k=1}^K \min \left( \rho_k A_k, \text{clip}(\rho_k, 1 - \epsilon, 1 + \epsilon) A_k \right) \right], \tag{27}$$

where $\rho_k = \frac{\pi_\phi(s_k \mid \mathbf{x}, \mathbf{s}_{1:k-1})}{\pi_\phi^{\text{OLD}}(s_k \mid \mathbf{x}, \mathbf{s}_{1:k-1})}$ is the importance ratio, and $\epsilon$ controls the trust region. This formulation prevents large updates by clipping the policy ratio.

Concretely, at each iteration, we collect 1,024 rollout steps with the frozen old policy $\pi_\phi^{\text{OLD}}$ (step-level, not episode-level). Each step stores the transition tuple $(\mathbf{x}, \mathbf{s}_{1:k-1}, s_k, R_k, \log \pi_\phi^{\text{OLD}}(s_k \mid \mathbf{x}, \mathbf{s}_{1:k-1}), V_\psi(\mathbf{x}, \mathbf{s}_{1:k-1}), \text{done})$ in a FIFO buffer of capacity 20,000 steps; once full, the oldest steps are discarded. we sample a mini-batch from the shuffled buffer and run PPO for 4 epochs. One epoch is completed when 1,024 sampled steps are used once for updates. After the update phase, we set $\pi_\phi^{\text{OLD}} \leftarrow \pi_\phi$ and repeat: collect the next 1,024 *steps*, push them into the buffer, and perform 4 epoch updates over those steps. This step-level sampling/updating schedule makes the rollout size and the optimization loop explicit.

In our implementation, we use $\epsilon = 0.2$ for trust region, $\gamma = 0.99$ for the discount factor, and $\beta = 0.01$ for entropy regularization coefficient which is applied to encourage exploration of diverse segmentations. All hyperparameters are used within recommended in prior PPO researches.

### A.3 COMBINATORIAL COMPLEXITY OF SEGMENT CANDIDATES

We prove that the number of candidate segment sets grows exponentially with the sequence length $T$. Throughout, we consider non-overlapping, non-empty discrete segments on indices $\{1, \ldots, T\}$. A single segment is specified by a start–end pair $(t^s, t^e)$ with $1 \leq t^s \leq t^e \leq T$, and a set of $K$ segments $\mathbf{s}_{1:K} = (s_1, \ldots, s_K)$ is required to be pairwise disjoint.

**Counting $K$ Disjoint Segments.** Fix $T \in \mathbb{N}$ and $K \in \{1, \ldots, \lfloor (T+1)/2 \rfloor\}$. Think of the time steps $\{1, \ldots, T\}$ as being separated by $T+1$ boundaries at positions $\{0, 1, \ldots, T\}$. Here, 0 denotes the position immediately to the left of the first time step, and $T$ denotes the position immediately to the right of the last time step. Picking a non-overlapping segment is the same as picking two boundaries and filling in everything between them; picking $K$ disjoint segments is the same idea repeated $K$ times, so in total, pick $2K$ distinct boundaries and pair them up in order: $(b_1, b_2)$, $(b_3, b_4)$, $\ldots$, $(b_{2K-1}, b_{2K})$. Each pair $(b_{2i-1}, b_{2i})$ forms a non-empty interval (since $b_{2i-1} < b_{2i}$), and the ordering prevents overlaps. Hence, choosing $K$ disjoint segments is equivalent to choosing $2K$ distinct boundaries out of $T+1$ possibilities, which gives $\binom{T+1}{2K}$.

**Total Number of Candidate Segment Sets.** Summing over all feasible $K$, the total number of non-empty segment sets is

$$\sum_{K=1}^{\lfloor (T+1)/2 \rfloor} \binom{T+1}{2K} = 2^T - 1 = \mathcal{O}(2^T).$$

Here $K$ denotes the number of disjoint segments selected. Each segment must contain at least one time step, so allocating $K$ segments requires at least $2K$ distinct boundaries (start and end). Since there are only $T+1$ available boundaries, the maximum feasible number of segments is $\lfloor (T+1)/2 \rfloor$.

**Remarks.** (i) This counting assumes segments are non-overlapping and non-empty; if overlaps or empty segments were allowed, the number of feasible segment sets would be larger. (ii) The exponential growth $2^T - 1$ shows how quickly the search space expands as $T$ grows, underscoring the combinatorial intractability of directly solving the joint optimization in Eq. (1). This motivates our decomposition into sequential CMI terms and policy-based optimization in the main paper.

## B PSEUDO CODE

---

**Algorithm 1** COLLECT (Step-level rollout with CMI reward and adaptive termination)

---

**Input:** Frozen old policy $\pi_\phi^{\text{OLD}}$, value $V_\psi$, classifier $g_\theta$, mask function $\text{MASK}(\cdot, \cdot)$, sparsity weights $\lambda$, termination $(\tau, K_{\max})$, rollout length $N_{\text{roll}}$

**Output:** Replay buffer $\mathcal{B}$ of step-level transitions

1: $\mathcal{B} \leftarrow \emptyset$
2: **while** $|\mathcal{B}| < N_{\text{roll}}$ **do**
3:     Sample minibatch $\mathbf{x}$ from training data
4:     $k \leftarrow 0, \mathbf{s}_{1:k} \leftarrow \emptyset, \texttt{done} \leftarrow \texttt{False}$
5:     $\mathbf{m}_{1:k} \leftarrow \mathbf{0}$                              ▷ zero vector mask (no segment yet)
6:     **while not** $\texttt{done}$ **and** $k < K_{\max}$ **do**
7:         $t_{k+1}^s \sim \pi_{\phi^s}^{\text{OLD}}(\cdot \mid \mathbf{x}, \mathbf{s}_{1:k}), \quad t_{k+1}^e \sim \pi_{\phi^e}^{\text{OLD}}(\cdot \mid t_{k+1}^s, \mathbf{x}, \mathbf{s}_{1:k})$     ▷ propose next segment
8:         $s_{k+1} = (t_{k+1}^s, t_{k+1}^e)$                              ▷ propose next segment
9:         $\mathbf{m}_{k+1} \leftarrow \text{SEGMENTTOMASK}(s_{k+1}); \quad \mathbf{m}_{1:k+1} \leftarrow \mathbf{m}_{1:k} \vee \mathbf{m}_k$
10:         $\mathbf{x}_{\mathbf{s}_{1:k}} \leftarrow \text{MASKING}(\mathbf{x}, \mathbf{m}_{1:k}); \quad \mathbf{x}_{\mathbf{s}_{1:k+1}} \leftarrow \text{MASKING}(\mathbf{x}, \mathbf{m}_{1:k+1})$
11:         $p_\theta(y \mid \mathbf{x}_{\mathbf{s}_{1:k}}) \leftarrow \text{softmax}(g_\theta(\mathbf{x}_{\mathbf{s}_{1:k}})); \quad p_\theta(y \mid \mathbf{x}_{\mathbf{s}_{1:k+1}}) \leftarrow \text{softmax}(g_\theta(\mathbf{x}_{\mathbf{s}_{1:k+1}}))$
12:         $r_\theta(s_{k+1}, \mathbf{s}_{1:k}) \leftarrow \mathbb{E}_{p_\theta(y|\mathbf{x})}\big[\log p_\theta(y \mid \mathbf{x}_{\mathbf{s}_{1:k}}) - \log p_\theta(y \mid \mathbf{x}_{\mathbf{s}_{1:k+1}})\big]$
13:         $c(s_{k+1}) \leftarrow \|\mathbf{m}_{k+1}\|_1 / T; \quad R_k \leftarrow r_\theta(s_{k+1}, \mathbf{s}_{1:k}) - \lambda c(s_{k+1})$
14:         $\texttt{done}_\tau \leftarrow 1 - \dfrac{\mathbb{E}_{p_\theta(y|\mathbf{x})}\big[\log p_\theta(y|\mathbf{x}_{\mathbf{s}_{1:k+1}})\big]}{\mathbb{E}_{p_\theta(y|\mathbf{x})}\big[\log p_\theta(y|\mathbf{x}_{\mathbf{s}_{1:k}})\big]} \geq \tau; \quad \texttt{done} \leftarrow \texttt{done}_\tau \textbf{ or } (k+1 \geq K_{\max})$
15:         $\log \pi^{\text{OLD}} \leftarrow \log \pi_{\phi^s}^{\text{OLD}}(t_{k+1}^s \mid \mathbf{x}, \mathbf{s}_{1:k}) + \log \pi_{\phi^e}^{\text{OLD}}(t_{k+1}^e \mid t_{k+1}^e, \mathbf{x}, \mathbf{s}_{1:k})$
16:         $v_{1:k} \leftarrow V_\psi(\mathbf{x}, \mathbf{s}_{1:k}); \quad v_{1:k+1} \leftarrow V_\psi(\mathbf{x}, \mathbf{s}_{1:k+1})$
17:         Push $\big(\mathbf{x}, \mathbf{s}_{1:k}, s_{k+1}, R_k, \log \pi^{\text{OLD}}, (v_{1:k}, v_{1:k+1}), \texttt{done}\big)$ into $\mathcal{B}$
18:         $\mathbf{m}_{1:k} \leftarrow \mathbf{m}_{1:k+1}; \quad k \leftarrow k + 1$
19:     **end while**
20: **end while**
21: **return** $\mathcal{B}$

---

**Algorithm 2** PPO_UPDATE (Clipped surrogate with step-level transitions)

---

**Input:** Buffer $\mathcal{B}$, current policy $\pi_\phi$, value $V_\psi$, epochs $E_{\text{ppo}}$, minibatch size $M$, clip $\epsilon$, discount $\gamma$, entropy coef $\beta$

**Output:** Updated $(\phi, \psi)$ and synced $\pi_\phi^{\text{OLD}}$

1: **PPO epochs**
2: **for** $e = 1$ to $E_{\text{ppo}}$ **do**
3:     **for all** minibatch $\mathcal{M}$ of size $M$ sampled from $\mathcal{B}$ **do**
4:         Unpack $\big(\mathbf{x}, \mathbf{s}_{1:k}, s_{k+1}, R_k, \log \pi^{\text{OLD}}, (v_{1:k}, v_{1:k+1}), \texttt{done}\big) \in \mathcal{M}$
5:         $\log \pi \leftarrow \log \pi_\phi(s_k \mid \mathbf{x}, \mathbf{s}_{1:k}); \quad \rho \leftarrow \exp(\log \pi^{\text{new}} - \log \pi^{\text{old}})$
6:         $A_k \leftarrow R_k + \gamma v_{1:k+} - v_{1:k}; A_k \leftarrow \text{sg}[A_k]$
7:         $L_{\text{clip}} \leftarrow \mathbb{E}\big[\min\big(\rho A_k, \text{clip}(\rho, 1 - \epsilon, 1 + \epsilon) A_k\big)\big]$
8:         $L_{\text{value}} \leftarrow \mathbb{E}\Big[\big(v_{1:k} - \text{sg}[R_k + \gamma v_{1:k+1}]\big)^2\Big]$
9:         $H \leftarrow \mathbb{E}\big[\text{Entropy}\big(\pi_\phi(\cdot \mid \mathbf{x}, \mathbf{s}_{1:k})\big)\big]$     ▷ $H$ means entropy of segment distribution
10:         $L_{\text{total}} \leftarrow -\big(L_{\text{clip}} + \beta H\big) + L_{\text{value}}$     ▷ maximize policy, minimize value loss
11:         Update $(\phi, \psi)$ by descending $\nabla L_{\text{total}}$
12:     **end for**
13: **end for**
14: **return** $(\phi, \psi, \pi_\phi^{\text{OLD}})$

---

**Remarks.** During inference, segments are deterministically sampled from the policy via mode values.

## C EXPERIMENT DETAILS

This appendix provides full details of datasets, preprocessing, model architectures, evaluation metrics, and ablations.

### C.1 DATASETS AND PREPROCESSING

#### C.1.1 SYNTHETIC DATASETS

To evaluate the ability of TimeSeg to recover ground-truth explanatory regions, we construct synthetic datasets where the causal subsequences are known by design. Following prior work (Queen et al., 2023; Liu et al., 2024), we embed class-defining patterns into a noisy background such that predictions rely only on the inserted segments. Each dataset is initialized with a non-autoregressive moving average (NARMA) noise base, into which task-specific motifs are inserted. This setup ensures that classification cannot be solved by shortcuts, and that faithful explanations must identify the correct subsequences. A summary of dataset statistics is provided in Table 5.

**SeqComb-UV.** Each univariate sequence of length 200 is constructed by inserting two non-overlapping subsequences chosen from an *increasing* trend (I) and a *decreasing* trend (D). Trends are generated using sinusoidal patterns with random wavelength and additive Gaussian noise. There are four classes: (i) null (no pattern), (ii) I,I, (iii) D,D, and (iv) I,D. Ground-truth explanations are the positions of the inserted I/D subsequences.

**FreqShapes-V.** We adapt the frequency-based synthetic dataset from (Queen et al., 2023). Each sequence has length 50, and the class label is determined by the periodicity of a spike pattern. We use two spike shapes (upward and downward) and two frequencies (periods of 10 and 19 time steps), forming four classes in total. The ground-truth explanatory regions are the positions of the inserted spikes. Unlike the original version, we remove duplicate motif occurrences so that each sequence contains a single definitive explanatory segment. See Figure 4 for a visual comparison between the original FreqShapes and our modified FreqShapes-V.

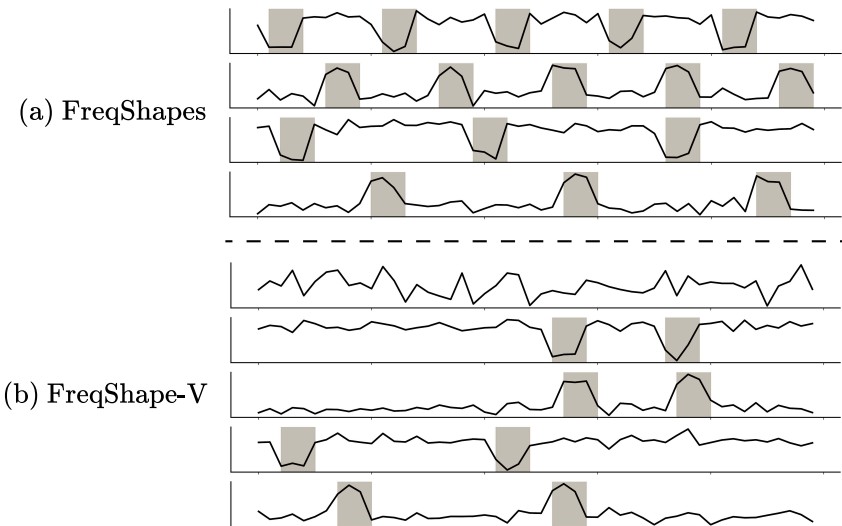

**Figure 4: Illustration of the difference between FreqShape and FreqShapes-V.**

**LowVarDetect-UV.** Each univariate sequence of length 200 is generated by inserting a low-variance region into the NARMA background. The predictive signal comes from both the variance level and the mean of the Gaussian noise in this subsequence. For binary classification, the low-variance subsequence is generated with either negative or positive mean, determining the label. This dataset differs from the anomaly-style tasks above, as the explanatory subsequence does not correspond to large amplitude changes but rather to more subtle statistical properties (variance reduction). Ground-truth explanations are the positions of the low-variance subsequences.

**Table 5:** Synthetic Dataset Description.

| Dataset | #Samples | Length | Dim | Classes |
|---|---|---|---|---|
| SeqComb-UV | 6,100 | 200 | 1 | 4 |
| FreqShapes-V | 6,100 | 50 | 1 | 5 |
| LowVarDetect-UV | 6,100 | 200 | 1 | 2 |

### C.1.2 REAL-WORLD DATASETS

We further evaluate TimeSeg on a suite of widely used real-world time-series classification benchmarks. Unlike the synthetic datasets, real-world tasks pose additional challenges due to noise, inter-subject variability, and the absence of segment-wise ground truth in most cases. Below we describe each dataset and, when available, the annotation used as segment-wise ground-truth explanatory regions.

**MIT-ECG.** We use the arrhythmia detection dataset from the MIT-BIH Arrhythmia Database (Moody & Mark, 2001), which provides electrocardiogram (ECG) recordings sampled at 360 Hz. Following common practice, we segment the recordings into short windows centered on individual beats. We focus on three representative classes: *Normal* (N), *Left Bundle Branch Block* (L), and *Right Bundle Branch Block* (R). Because both L and R diagnoses are known to rely on the morphology of the QRS complex, we use the cardiologist-annotated QRS intervals as segment-wise ground-truth explanatory regions. This dataset allows quantitative evaluation of explanation overlap against clinically annotated ground truth. For comparability, we follow the exact preprocessing protocol of TimeX (Queen et al., 2023).

**Epilepsy.** The Epileptic Seizure Recognition dataset (Andrzejak et al., 2002) contains EEG recordings from 500 subjects. Each subject's brain activity was recorded for 23.6 seconds, which was then partitioned into non-overlapping 1-second windows, resulting in 11,500 single-channel sequences of length 178 sampled at 178 Hz. The dataset provides five labels: (i) eyes open, (ii) eyes closed, (iii) EEG from a healthy brain region, (iv) EEG from a tumor region, and (v) seizure. For binary seizure detection, we merge the first four classes into the negative class and retain seizure as the positive class. No segment-wise ground-truth annotations are available; thus evaluation is based on explanation fidelity metrics (sufficiency/comprehensiveness). We also adopt the identical preprocessing as TimeX (Queen et al., 2023).

**Wafer.** The Wafer dataset (Dau et al., 2019) is derived from semiconductor manufacturing processes, where sensor signals are used to detect faults in wafer production. It contains univariate time-series of varying lengths that are standardized to a common length for evaluation. The classification task is to distinguish between normal and abnormal wafers. Since no segment-wise ground truth is provided, we rely on fidelity and structural quality metrics to assess explanations.

**GunPoint.** The GunPoint dataset (Dau et al., 2019) consists of motion capture time-series that record hand movements. Subjects perform two distinct gestures: drawing a gun from a holster versus pointing with a finger. Each sequence is univariate with length 150. Prior analyses of this dataset suggest that informative patterns often coincide with short transitional movements (e.g., raising or lowering) rather than the static middle portion, though this can vary across instances. Accordingly, GunPoint provides a convenient testbed to qualitatively assess whether an explainer highlights plausible transition segments as segment-wise explanatory regions.

**Table 6:** Real-World Dataset Description.

| Dataset | #Samples | Length | Dim | Classes |
|---|---|---|---|---|
| MIT-ECG | 90,337 | 360 | 1 | 2 |
| Epilepsy | 11,500 | 178 | 1 | 2 |
| Wafer | 7,164 | 152 | 1 | 2 |
| GunPoint | 400 | 150 | 1 | 2 |

## C.2 EVALUATION METRICS

To assess the quality of segment-wise explanations, we employ three complementary categories of metrics: **Overlap with Ground Truth**, **Explanation Fidelity**, and **Segment Quality**. These metrics enable both quantitative comparisons on datasets with annotated explanatory regions and qualitative assessment on datasets without annotations.

**Overlap with Ground Truth.** For datasets with segment-wise ground-truth annotations (e.g., synthetic datasets and MIT-ECG), we measure the degree of overlap between the predicted explanation mask $\mathbf{m}_{1:K}$ and the ground-truth mask $\mathbf{m}^\star$. Specifically, we report the **F1-score** and the **Jaccard index (IoU)**:

$$\text{F1} = \frac{2 \cdot |\mathbf{m}_{1:K} \cap \mathbf{m}^\star|}{|\mathbf{m}_{1:K}| + |\mathbf{m}^\star|}, \quad \text{IoU} = \frac{|\mathbf{m}_{1:K} \cap \mathbf{m}^\star|}{|\mathbf{m}_{1:K} \cup \mathbf{m}^\star|}.$$

**Explanation Fidelity.** To quantify how well the identified segments capture the predictive signal used by the black-box model, we follow the insertion/removal evaluation framework:

$$\text{SUFF}{\downarrow} = \frac{\text{AUROC}_{\text{base}} - \text{AUROC}_{\text{insert}}}{\text{AUROC}_{\text{base}}}, \quad \text{COMP}{\uparrow} = \frac{\text{AUROC}_{\text{base}} - \text{AUROC}_{\text{remove}}}{\text{AUROC}_{\text{base}}}.$$

Here $\text{AUROC}_{\text{base}}$ denotes the AUROC on the original time-series, $\text{AUROC}_{\text{insert}}$ the AUROC when only the selected segments are retained, and $\text{AUROC}_{\text{remove}}$ the AUROC when the selected segments are removed. Masked regions are replaced with either zeros or the dataset mean. Lower SUFFI-CIENCY and higher COMPREHENSIVENESS indicate more faithful explanations.

**Segment Quality.** Beyond fidelity, we evaluate the structural properties of the produced explanations:

- **Sparsity** ($\downarrow$): the fraction of time points selected, $\text{Sparsity} = \frac{1}{T}\|\mathbf{m}_{1:K}\|_1$. Lower values indicate more concise explanations.
- **Contiguity** ($\downarrow$): the normalized number of transitions between selected and unselected points,

$$\text{Contiguity} = \frac{1}{T-1} \sum_{t=1}^{T-1} \mathbb{I}[m_{1:K,t} \neq m_{1:K,t+1}],$$

where smaller values indicate fewer boundaries, meaning more coherent segment-wise patterns.

## C.3 IMPLEMENTATION DETAILS

We provide implementation details of the black-box predictor, the proposed segment-wise explainer, and the reinforcement learning optimization, as well as hardware and reproducibility information. For completeness and reproducibility, we summarize all key hyperparameters used in our experiments in Table 8.

**Black-box Predictor.** For all datasets, the target black-box model $g_\theta$ is implemented as a Temporal Convolutional Network (TCN) (Lea et al., 2017). The network consists of 6 convolutional blocks with kernel size 3 and exponentially increasing dilations, each followed by residual connections, ReLU activation, and dropout of $p = 0.1$. Global average pooling and a fully-connected classifier map to the label space. We train $g_\theta$ using cross-entropy loss with Adam optimizer (learning rate $10^{-3}$, weight decay $10^{-4}$, batch size 128). Early stopping is applied based on validation F1 with patience of 25 epochs.

**Explainer Networks (Policy and Value).** At each decision step $k$, both the policy $\pi_\phi$ and the value network $V_\psi$ take as input the full sequence $\mathbf{x}$ together with the current explanation mask $\mathbf{m}_{1:k-1}$, concatenated along the feature axis to form $[\mathbf{x}; \mathbf{m}_{1:k-1}] \in \mathbb{R}^{T \times (C+1)}$. This design provides the global temporal context while explicitly encoding which regions have already been selected.

The policy is factorized into a start-policy $\pi_{\phi^s}$ and an end-policy $\pi_{\phi^e}$, and each uses its own encoder backbone. To inject explicit dependency between the start and end distributions, the pooled embedding produced by $\pi_{\phi^s}$ is added to the pooled embedding of $\pi_{\phi^e}$ before the final linear layer of $\pi_{\phi^e}$. In this way, the choice of an end index is conditioned on both the input sequence and the representation of the chosen start index.

**Table 7:** AUROC performance of the black-box predictor $g_\theta$ on each dataset.

| Dataset | #Classes | AUROC |
|---|---|---|
| SeqComb-UV | 4 | $0.99_{\pm 0.00}$ |
| FreqShapes-V | 5 | $1.00_{\pm 0.00}$ |
| LowVarDetect-UV | 2 | $0.99_{\pm 0.00}$ |
| MIT-ECG | 2 | $0.99_{\pm 0.00}$ |
| Epilepsy | 2 | $0.99_{\pm 0.00}$ |
| Wafer | 2 | $0.99_{\pm 0.00}$ |
| GunPoint | 2 | $0.99_{\pm 0.00}$ |

Concretely, after sampling a start index $t^s \sim \pi_{\phi^s}(\cdot \mid \mathbf{x}, \mathbf{s}_{1:k-1})$, we enforce the ordering constraint $t^e \geq t^s$ by *masking* the end-policy's categorical distribution before the final softmax. the end point probability $\pi_{\phi^e}(t^e \mid t^s, \cdot) = 0$ for all $t^e < t^s$ and is renormalized over $\{t^s, \ldots, T\}$, which guarantees $t^e \geq t^s$ by construction. In practice, the mask is implemented by adding $-\infty$ (or a large negative constant) to the logits at indices $t < t^s$ before the softmax. This start-conditioned masking is applied at every decision step $k$.

The value network $V_\psi$ also consumes $[\mathbf{x}; \mathbf{m}_{1:k-1}]$ as input, with its own encoder backbone that may differ from both $\pi_{\phi^s}$ and $\pi_{\phi^e}$. It outputs a scalar state-value estimate, serving as the critic in the actor–critic framework.

**Table 8:** Hyperparameters used for TimeSeg.

| | |
|---|---|
| $N_{\text{buffer}}$ | 4,096 |
| $N_{\text{rollout}}$ | 1,024 |
| Batch size | 256 |
| Total epochs | 600 |
| PPO epochs | 4 |
| Optimizer | AdamW |
| Policy LR | $1 \times 10^{-4}$ |
| Value LR | $1 \times 10^{-4}$ |
| Weight decay (value) | $1 \times 10^{-4}$ |
| Cosine LR scheduler | True |
| Warmup epochs | 100 |
| Warmup start LR | $3.3 \times 10^{-5}$ |
| Policy final LR | $5 \times 10^{-5}$ |
| Value final LR | $2 \times 10^{-5}$ |
| CNN layers (policy/value) | 3 |
| CNN kernel size | 3 |
| Hidden dimension | 128 |
| MLP layers | 2 |
| $\gamma$ (discount factor) | 0.99 |
| PPO clip ratio $\epsilon$ | 0.2 |
| PPO entropy coefficient | 0.01 |
| Value loss coefficient | 1.0 |
| $\lambda$ (length penalty) | 0.3, 0.5 |
| $\tau$ (termination threshold) | 0.3 |
| $K_{\text{max}}$ | 5 |

# D  ADDITIONAL RESULTS

## D.1  SYNTHETIC PREDICTION PERFORMANCE

We report the prediction performance of the black-box classifier $g_\theta$ on the synthetic benchmarks.

**Table 9:** Fidelity on synthetic datasets (single row with three dataset blocks; continuity removed).

| Method | SeqComb-UV | | | | FreqShapes-V | | | | LowVarDetect-UV | | | |
|---|---|---|---|---|---|---|---|---|---|---|---|---|
| | Avg. | | Zero | | Avg. | | Zero | | Avg. | | Zero | |
| | Suff. | Comp. | Suff. | Comp. | Suff. | Comp. | Suff. | Comp. | Suff. | Comp. | Suff. | Comp. |
| Random | $18.43_{\pm1.19}$ | $0.08_{\pm0.04}$ | $47.41_{\pm3.20}$ | $0.20_{\pm0.11}$ | $8.59_{\pm2.05}$ | $0.00_{\pm0.00}$ | $30.31_{\pm5.99}$ | $0.00_{\pm0.00}$ | $28.61_{\pm2.90}$ | $0.53_{\pm0.08}$ | $45.17_{\pm1.68}$ | $1.39_{\pm0.18}$ |
| WinIT | $\underline{5.14}_{\pm2.47}$ | $4.47_{\pm0.57}$ | $32.18_{\pm2.55}$ | $5.66_{\pm0.93}$ | $0.40_{\pm0.50}$ | $0.57_{\pm0.689}$ | $\underline{0.55}_{\pm0.65}$ | $0.99_{\pm1.07}$ | $\underline{3.24}_{\pm1.19}$ | $13.84_{\pm3.55}$ | $\underline{0.98}_{\pm0.24}$ | $17.21_{\pm3.9}$ |
| Dynamask | $17.60_{\pm2.80}$ | $0.48_{\pm0.19}$ | $45.14_{\pm3.66}$ | $1.35_{\pm0.67}$ | $8.03_{\pm1.58}$ | $0.01_{\pm0.01}$ | $27.43_{\pm5.05}$ | $0.01_{\pm0.02}$ | $16.80_{\pm5.33}$ | $12.93_{\pm5.91}$ | $26.51_{\pm6.36}$ | $15.01_{\pm6.03}$ |
| TimeX++ | $7.25_{\pm4.45}$ | $\underline{23.99}_{\pm9.15}$ | $\textbf{14.35}_{\pm7.39}$ | $\underline{24.43}_{\pm9.65}$ | $0.22_{\pm0.32}$ | $\textbf{14.98}_{\pm17.42}$ | $\textbf{0.32}_{\pm0.28}$ | $\textbf{23.54}_{\pm16.08}$ | $7.01_{\pm3.21}$ | $\textbf{30.86}_{\pm11.69}$ | $6.67_{\pm5.14}$ | $\textbf{36.43}_{\pm11.21}$ |
| **TimeSeg (Ours)** | $\textbf{0.22}_{\pm0.15}$ | $\textbf{30.80}_{\pm1.60}$ | $\underline{28.71}_{\pm1.92}$ | $\textbf{26.77}_{\pm1.56}$ | $\textbf{0.00}_{\pm0.00}$ | $\underline{1.47}_{\pm0.58}$ | $1.21_{\pm1.71}$ | $2.91_{\pm0.98}$ | $\textbf{0.04}_{\pm0.040}$ | $\underline{24.76}_{\pm2.64}$ | $\textbf{0.47}_{\pm0.47}$ | $20.74_{\pm2.77}$ |

## D.2  ABLATION ON SEGMENT DISTRIBUTIONS (MIT-ECG)

We study how the choice of the segment-index distribution affects performance on **MIT-ECG**. Recall that our policy factorizes the segment into $s_k = (t_k^s, t_k^e)$ and enforces $t^e \geq t^s$ via start-conditioned masking (see Appendix C.3). Here, we replace the default distribution with several variants while keeping all other components identical to the main setting.

**Variants of the Policy Network.**

- *Cat–Cat* (default): start $\pi_{\phi^s}$ and end $\pi_{\phi^e}$ are categorical over absolute indices with start-conditioned masking.

- *Cat–Dur (Negative Binomial):* start categorical; end is parameterized as *duration* $d = t^e - t^s + 1$ via a Negative Binomial (NB); we map $(t^s, d) \mapsto t^e$ and mask invalid durations.

- *Cat–Cat (+CauchySmooth):* Cat–Cat with Cauchy smoothing on the end categorical distributon probabiliies $\pi_{\phi^e}$.

**Cauchy Smoothing.** Under the assumption that two nearby time points should be more similar than those far away by modeling durations, we smooth the probabilities by assigning higher weights to nearby indices using the Cauchy kernel

$$c_{\text{cauchy}}(\tau, \tau') = \sigma^2 \left(1 + \frac{(\tau - \tau')^2}{\ell^2}\right)^{-1},$$

which can be seen as a mixture of infinitely many RBF kernels with different length scales. As $\ell \to \infty$, $c_{\text{cauchy}}(\tau, \tau') \to \sigma^2$, recovering uniform weights.

**Table 10:** Performance comparison on **MIT-ECG** at an equal masking ratio (sparsity = 0.34) using the *Cat–Cat (+ CauchySmooth)* segment distribution. We report mean $\pm$ std over five folds; **bold** is best, underlined is second-best.

| Method | Overlap | | Fidelity | | Continuity |
|---|---|---|---|---|---|
| | F1 | IoU | Suff. | Comp. | |
| Dynamask | $0.29_{\pm0.01}$ | $0.17_{\pm0.01}$ | $35.97_{\pm6.98}$ | $5.19_{\pm2.99}$ | $0.12_{\pm0.02}$ |
| TimeX++ | $\textbf{0.48}_{\pm0.12}$ | $\textbf{0.33}_{\pm0.10}$ | $\underline{21.43}_{\pm3.32}$ | $\textbf{19.05}_{\pm7.25}$ | $\underline{0.02}_{\pm0.00}$ |
| **TimeSeg (Ours)** | $\underline{0.46}_{\pm0.03}$ | $\underline{0.32}_{\pm0.03}$ | $\textbf{0.05}_{\pm0.03}$ | $\underline{7.69}_{\pm4.46}$ | $\textbf{0.01}_{\pm0.00}$ |

**Remarks.** Although modeling duration with a Negative Binomial distribution is conceptually attractive, it proved fragile in practice, as small changes in parameterization $(r, p)$ led to large shifts in the implied length prior. Consequently, we could not obtain stable gains across experiments. Cauchy smoothing increases the entropy of the end-policy logits, which reduces run-to-run variance but does not necessarily yield the best overlap/fidelity trade-off.

### D.3 QUALITATIVE EXAMPLES

To complement the quantitative results, we provide qualitative comparisons of explanations generated by LIMESegment, TimeX++, and TimeSeg on two representative datasets: **SeqComb-UV** (synthetic) and **MIT-ECG** (real-world).

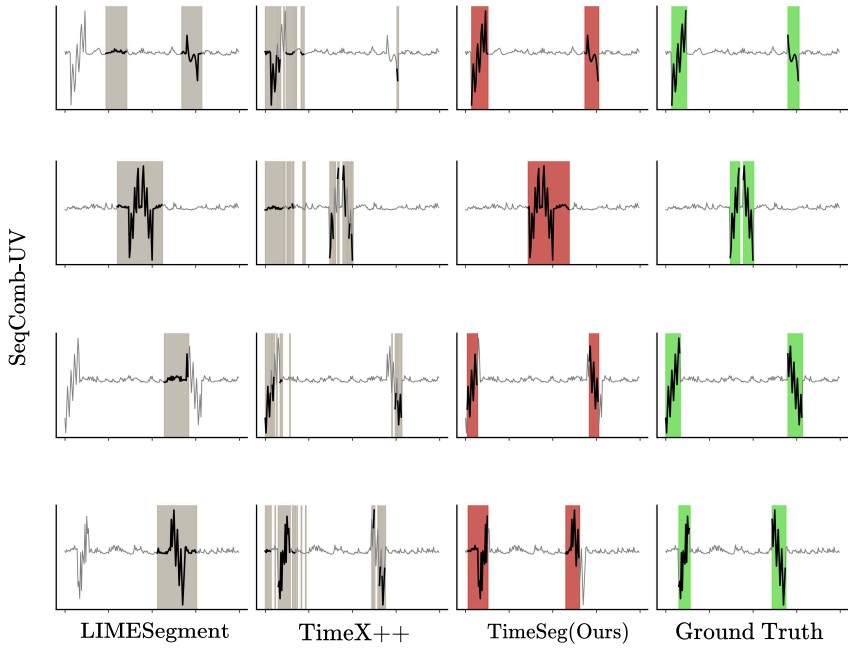

**Figure 5: Qualitative examples on *SeqComb-UV*.** Each row shows a test instance with explanations from LIMESEGMENT, TIMEX++, and TimeSeg, followed by the ground-truth segment (green).

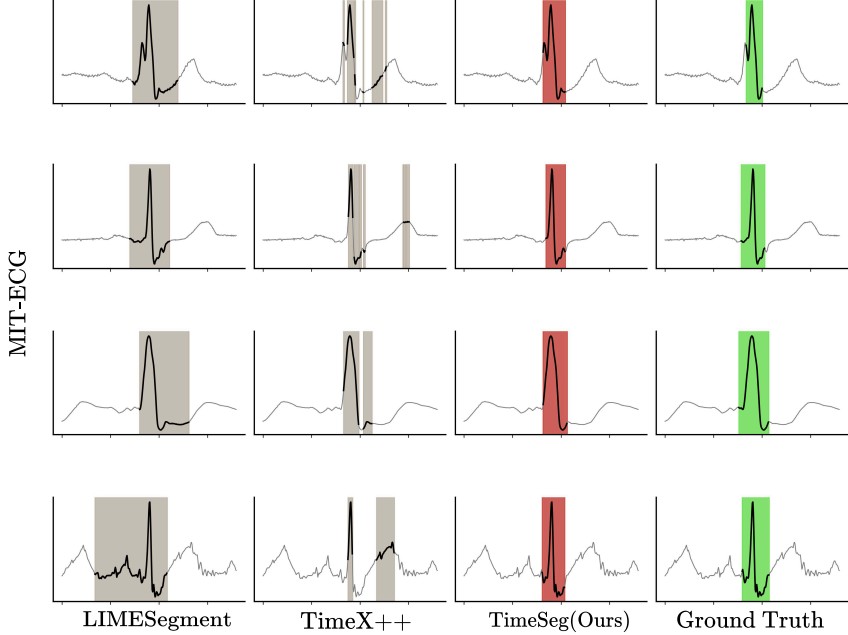

**Figure 6: Qualitative examples on *MIT-ECG*.** Columns show explanations from LIMESegment, TimeX++, and TimeSeg, with the cardiologist-annotated QRS interval as ground truth (green). LIMESegment remains contiguous but can be misaligned, TimeX++ highlights scattered points without clear segment structure, and TimeSeg retrieves coherent segments closely matching the QRS complex boundaries.

## D.4 SENSITIVITY ANALYSIS ON $\tau$

We analyze the effect of the termination threshold $\tau$, which governs *how early* the policy stops selecting new segments, and thus directly influences the number and structure of the selected segments.

The effective number of segments is instead primarily determined by $\tau$ (termination) and jointly by $\lambda$ (length penalty). We therefore report a sensitivity analysis over $\tau$ on **SeqComb-UV**, measuring how sparsity, overlap quality (F1/IoU), contiguity, and the selected segment structure change under different values.

**Effect of $\tau$.** Smaller values of $\tau$ make it easier for the policy to keep adding segments, so it tends to select *more* segments per instance. Under the sparsity penalty, this pressure is absorbed by shortening individual segments, leading to many short segments and slightly higher overall sparsity. As $\tau$ increases, the policy stops earlier, selects *fewer* segments, and instead allocates longer duration to each segment to match the sparsity budget. In our experiments on **SeqComb-UV**, F1/IoU remains stable over a wide range of $\tau$ (*e.g.*, 0.1–0.5), while non-positive $\tau$ values (*i.e.*, 0 and $-0.1$) effectively provide a margin that allows the policy to add new segments even when the immediate information gain is small, resulting in more frequent segment selections.

Table 11 reports the full quantitative results.

**Table 11:** Sensitivity to termination threshold $\tau$.

| $\tau$ | Sparsity | F1 | IoU | Contiguity | # Segments | Segment Length |
|---|---|---|---|---|---|---|
| -0.1 | $0.170 \pm 0.037$ | $0.523 \pm 0.031$ | $0.377 \pm 0.030$ | $0.025 \pm 0.002$ | $4.747 \pm 0.038$ | $10.219 \pm 2.280$ |
| 0 | $0.193 \pm 0.029$ | $0.627 \pm 0.026$ | $0.468 \pm 0.029$ | $0.022 \pm 0.001$ | $3.785 \pm 0.059$ | $16.998 \pm 2.186$ |
| 0.1 | $0.143 \pm 0.021$ | $0.667 \pm 0.024$ | $0.524 \pm 0.022$ | $0.019 \pm 0.001$ | $1.865 \pm 0.057$ | $16.607 \pm 2.509$ |
| 0.3 | $0.141 \pm 0.016$ | $0.645 \pm 0.018$ | $0.497 \pm 0.018$ | $0.017 \pm 0.000$ | $1.631 \pm 0.060$ | $17.615 \pm 2.869$ |
| 0.5 | $0.137 \pm 0.030$ | $0.618 \pm 0.012$ | $0.468 \pm 0.009$ | $0.015 \pm 0.000$ | $1.521 \pm 0.040$ | $16.364 \pm 2.265$ |
| 0.7 | $0.138 \pm 0.031$ | $0.612 \pm 0.018$ | $0.459 \pm 0.020$ | $0.014 \pm 0.000$ | $1.419 \pm 0.021$ | $18.070 \pm 2.873$ |
| 0.9 | $0.136 \pm 0.020$ | $0.599 \pm 0.013$ | $0.443 \pm 0.014$ | $0.013 \pm 0.000$ | $1.300 \pm 0.031$ | $19.487 \pm 2.370$ |

## D.5 MULTIVARIATE TIME-SERIES EXTENSION

**Why we have focused on a Univariate Setup.** In the main text we deliberately focused on *univariate* time series in order to highlight our primary contribution: moving from point-wise to segment-wise explanations under strict black-box constraints, and making the resulting combinatorial search over variable-length segments tractable. In this appendix section, we describe how the same formulation can be *naturally* extended to the multivariate setting by adding a simple channel-selection component to the action space, without modifying the underlying reward or black-box model.

**Simple multivariate extension.** To extend TimeSeg to multivariate inputs, we augment the action space with a **channel-selection policy**. Each segment becomes a joint draw

$$\mathbf{s} = (c, t^s, t^e), \qquad c \in [d],$$

and the policy factorizes as

$$\pi_\phi(\mathbf{s} \mid \cdot) = \pi_\phi(c, t^s, t^e \mid \cdot) = \pi_{\phi^c}(c \mid \cdot)\, \pi_{\phi^s}(t^s \mid c, \cdot)\, \pi_{\phi^e}(t^e \mid c, t^s, \cdot).$$

Here, $c$ denotes the chosen *channel index*, and $\pi_{\phi^c}$ is a categorical distribution whose logits are produced by the policy network. All other components – the reward, sparsity penalty, termination, and black-box predictor – remain unchanged; only the action space is expanded.

**Additional experiments.** We evaluate the multivariate version of TimeSeg on **SeqComb-MV**, a multivariate extension of **SeqCombSingle** in which channels are constructed with minimal cross-channel interaction. TimeSeg achieves **comparable F1/IoU** to the univariate setting while preserving markedly better contiguity (Table 12), indicating that the framework transfers with minimal modification.

**Table 12:** Multivariate results on SeqComb-MV (sparsity: 0.063).

| Method | F1 | IoU | Contiguity |
|---|---|---|---|
| TimeX++ | $0.471_{\pm 0.038}$ | $0.334_{\pm 0.030}$ | $0.064_{\pm 0.009}$ |
| Ours (TimeSeg) | $\mathbf{0.502}_{\pm \mathbf{0.034}}$ | $\mathbf{0.353}_{\pm \mathbf{0.032}}$ | $\mathbf{0.003}_{\pm \mathbf{0.000}}$ |

## D.6 BLACK-BOX ARCHITECTURES: GENERALIZATION

Beyond TCN, we assess whether TimeSeg transfers to qualitatively different black-box architectures. To this end, we additionally evaluate **RNN** and **Transformer** classifiers on three representative datasets: **SeqCombSingle** (synthetic), **MIT-ECG** (real, with ground-truth QRS annotations), and **GunPoint** (real, without segment annotations), with results reported in Tables 13 to 18.

Across all architectures and datasets, TimeSeg exhibits the *same qualitative behavior*: (1) compact and coherent segments, (2) stable overlap when ground truth is available, (3) consistently strong sufficiency/comprehensiveness fidelity, (4) minimal dependence on the specific backbone. This supports that TimeSeg operates effectively under a strict black-box setting without relying on gradient access or model structure, as summarized in Figure 7.

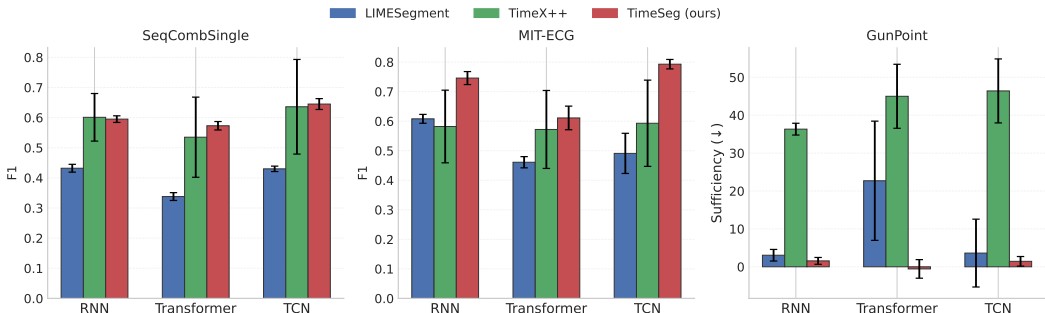

**Figure 7: Backbone generalization.** For each dataset (SeqCombSingle, MIT-ECG, GunPoint), we compare LIMESegment, TimeX++, and TimeSeg across three backbones (TCN, RNN, Transformer). Left/middle: F1 overlap with ground-truth explanatory regions (higher is better). Right: sufficiency on GunPoint (lower is better). TimeSeg matches or exceeds the best performance across backbones while keeping the standard deviation consistently small, indicating that its selected segments capture most of the predictive signal regardless of the underlying architecture.

**RNN results.** On SeqCombSingle and MIT-ECG, TimeSeg matches or exceeds TimeX++ on IoU/F1 while producing far more contiguous explanations. On GunPoint, TimeSeg achieves the lowest sufficiency (best) among all methods except LIMESegment, while still capturing coherent segments rather than scattered points.

**Table 13:** RNN backbone: SeqCombSingle (sparsity: 0.24).

| Method | F1 | IoU | Contiguity |
|---|---|---|---|
| IG | $0.515_{\pm 0.028}$ | $0.352_{\pm 0.026}$ | $0.113_{\pm 0.014}$ |
| LIMESegment | $0.432_{\pm 0.013}$ | $0.290_{\pm 0.011}$ | $\mathbf{0.011}_{\pm \mathbf{0.000}}$ |
| TimeX++ | $\mathbf{0.601}_{\pm \mathbf{0.079}}$ | $0.434_{\pm 0.069}$ | $0.141_{\pm 0.004}$ |
| **Ours** | $0.595_{\pm 0.011}$ | $\mathbf{0.437}_{\pm \mathbf{0.012}}$ | $0.012_{\pm 0.000}$ |

**Table 14:** RNN backbone: MIT-ECG (sparsity: 0.12).

| Method | F1 | IoU | Contiguity |
|---|---|---|---|
| IG | 0.628±0.016 | 0.477±0.018 | 0.035±0.001 |
| LIMESegment | 0.608±0.015 | 0.476±0.015 | **0.006±0.000** |
| TimeX++ | 0.582±0.123 | 0.463±0.106 | 0.013±0.002 |
| **Ours** | **0.746±0.022** | **0.628±0.026** | **0.006±0.000** |

**Table 15:** RNN backbone: GunPoint (sparsity: 0.06).

| Method | Suff. (Avg.) | Comp. (Avg.) | Suff. (Zero) | Comp. (Zero) | Contiguity |
|---|---|---|---|---|---|
| Random | 55.029±2.799 | 3.572±3.708 | 55.043±2.817 | 3.572±3.708 | 0.114±0.061 |
| IG | 34.784±9.174 | 18.223±10.334 | 34.784±9.174 | 18.223±10.334 | 0.027±0.007 |
| LIMESegment | 3.049±1.533 | **65.196±9.383** | 3.049±1.533 | **65.196±9.383** | 0.009±0.001 |
| TimeX++ | 36.322±1.555 | 1.465±2.112 | 36.318±1.556 | 1.465±2.112 | **0.008±0.001** |
| **Ours** | **1.557±0.903** | 26.688±17.553 | **1.557±0.903** | 26.688±17.553 | 0.012±0.001 |

**Transformer results.** We repeat the same experiment using a Transformer encoder as the black-box. TimeSeg again achieves the best or second-best overlap on SeqCombSingle and MIT-ECG while maintaining very low contiguity, and exhibits strong sufficiency and comprehensiveness behavior on GunPoint.

**Table 16:** Transformer backbone: SeqCombSingle (sparsity: 0.25).

| Method | F1 | IoU | Contiguity |
|---|---|---|---|
| IG | 0.471±0.028 | 0.314±0.025 | 0.172±0.010 |
| LIMESegment | 0.338±0.013 | 0.215±0.010 | **0.011±0.000** |
| TimeX++ | 0.535±0.133 | 0.371±0.090 | 0.093±0.009 |
| **Ours** | **0.573±0.014** | **0.416±0.014** | 0.012±0.000 |

**Table 17:** Transformer backbone: MIT-ECG (sparsity: 0.13).

| Method | F1 | IoU | Contiguity |
|---|---|---|---|
| IG | 0.462±0.019 | 0.313±0.017 | 0.102±0.016 |
| LIMESegment | 0.461±0.041 | 0.343±0.044 | 0.007±0.001 |
| TimeX++ | 0.572±0.132 | 0.463±0.098 | 0.013±0.003 |
| **Ours** | **0.611±0.040** | **0.484±0.037** | **0.006±0.000** |

**Table 18:** Transformer backbone: GunPoint (sparsity: 0.14).

| Method | Suff. (Avg.) | Comp. (Avg.) | Suff. (Zero) | Comp. (Zero) | Contiguity |
|---|---|---|---|---|---|
| Random | 48.152±11.295 | 8.601±3.745 | 48.152±11.295 | 8.601±3.745 | 0.242±0.049 |
| IG | 13.547±4.598 | 33.455±14.972 | 13.547±4.598 | 33.455±14.972 | 0.146±0.036 |
| LIMESegment | 22.703±15.732 | **68.148±14.702** | 22.703±15.732 | **68.148±14.702** | 0.014±0.003 |
| TimeX++ | 44.995±8.446 | 3.745±2.406 | 44.999±8.447 | 3.745±2.406 | **0.007±0.001** |
| **Ours** | **-0.563±2.452** | 40.953±7.352 | **-0.561±2.451** | 40.953±7.352 | 0.014±0.001 |

**Summary.** These results demonstrate that TimeSeg is architecture-agnostic, producing coherent, compact, and high-fidelity segment-wise explanations across convolutional, recurrent, and Transformer encoder backbones.

### D.7 COMPUTATIONAL COST

**Interactions.** Standard RL typically rolls out episodes for $\sim 10^3$–$10^4$ steps, incurring a large number of agent–environment interactions. In contrast, we cast RL as a *segment selection* problem: for

each input we take at most $K_{\max}$ decisions, and we explicitly cap the number of sampled steps per epoch (e.g., $N_{\text{rollout}} = 1024$). Collected trajectories are stored in a replay buffer and reused until eviction. As a result, our interaction budget is far below that of standard RL, and comparable to STE-based masking methods that also query the black box once per generated mask.

**Computational cost.** We report *training* wall-clock time per batch (batch size 256), averaged over 1,000 batches, and *inference* wall-clock time per 1,000 samples for each method (Table 19). For TimeSeg, the reported training time explicitly includes both the *segment-selection rollout* (policy interaction with the black-box) and the ensuing PPO update steps. LIMESegment is training-free, so only inference time is reported. All measurements were obtained on a same hardware setup with an Intel(R) Xeon(R) CPU and an NVIDIA RTX A6000 GPU, and all methods use identical preprocessing.

**Table 19:** Wall-clock cost on **SeqCombSingle**. Rollout and training times are per batch (bs = 256), averaged over 1,000 batches; inference times are per 1,000 samples.

| Method | Rollout Time (s) | Training Time (s) | Inference Time (s) |
|---|---|---|---|
| LIMESegment | – | – | $6.166\pm_{1.620}$ |
| TimeX++ | – | $0.250\pm_{0.025}$ | $0.003\pm_{0.000}$ |
| **Ours** | $0.267\pm_{0.059}$ | $0.040\pm_{0.075}$ | $0.023\pm_{0.036}$ |

**Efficiency, stability, convergence.** To make training behavior transparent, we report both *training* and *validation* reward curves, and we further decompose the validation reward into its cross-entropy and length (sparsity) components. This reveals how the sparsity regularizer shapes the learning dynamics and how quickly the policy stabilizes (Figure 8).

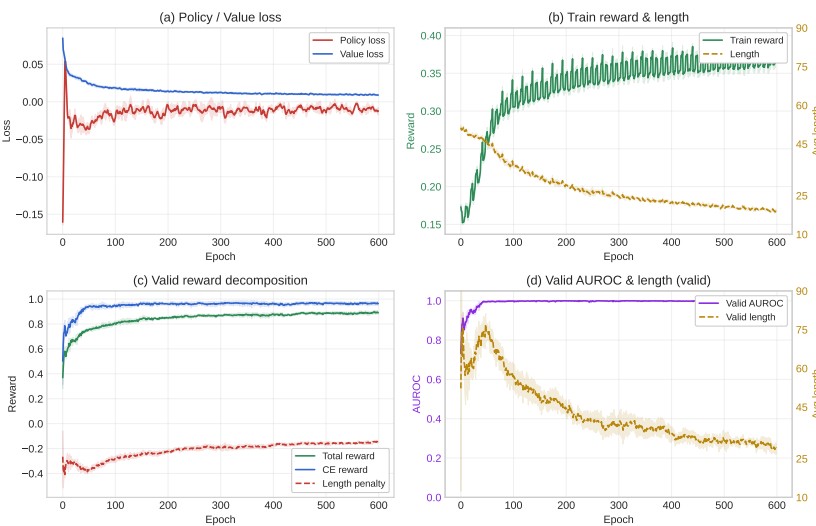

**Figure 8: Training dynamics on SeqCombSingle.** (a) policy/value losses; (b) training reward and average length; (c) validation total reward and its decomposition into cross-entropy and length terms; (d) validation AUROC and average explanation length. Curves show mean $\pm$ std over five folds.

## D.8  SEGMENT CLUSTERING ANALYSIS

To provide population-level evidence for this structural view, we perform a simple clustering analysis on the extracted segments, and visualize the results in Figure 9. Concretely, we:

1. collect all segments selected by TimeSeg on the test split of a given MIT-ECG dataset.
2. compute pairwise DTW distances between segments and obtain a 2D embedding via a distance-based method.

3. cluster the embedded segments with $k$-medoids using the DTW space.

4. visualize both the embedded points and the segments at the cluster centroid.

This yields groups of explanatory segments that correspond to recurring temporal segments (e.g., QRS-like shapes in **MIT-ECG**). Such segment-level structure is difficult to recover from purely point-wise saliency but becomes straightforward once explanations are expressed as contiguous segments as in our TimeSeg.

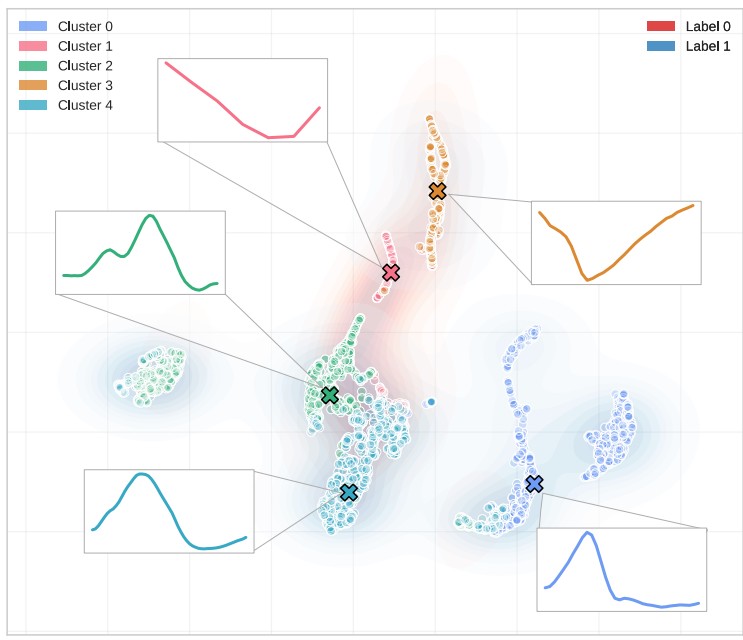

**Figure 9: Segment clustering on *MIT-ECG*.** Each point is an extracted segment embedded in a DTW-based space; colors denote clusters (via $k$-medoids), and crosses mark the cluster medoids. The corresponding medoid segments form representative QRS-like segments, illustrating that TimeSeg 's contiguous segments aggregate into clinically meaningful patterns at the population level.

## E    THE USE OF LLMS IN THIS WORK

LLMs were used solely for minor sentence editing to improve readability and flow, with no involvement in idea generation, experimental design, analysis, or substantive content creation.

