# OpenReview forum: "TimeSeg: An Information-Theoretic Segment-Wise Explainer for Time-Series Predictions"
_ICLR.cc/2026/Conference — ICLR 2026 Poster_

### Official Review · Reviewer_EPf4 · 2025-10-21

**Soundness:** 3
**Presentation:** 3
**Contribution:** 3
**Rating:** 6
**Confidence:** 2

**Summary:**

The paper introduces TimeSeg, an explanation method to highlight segments in a (univariate) time series that contributes highly to the prediction. The method is in contrast of the previous approaches that highlight individual observations in a time series. The method leverage the use of mutual information and reinforcement learning.

**Strengths:**

1) The paper is written quite well. The flow and logic are mostly clear and notations are good as well.
2) The premise of the paper is clear and reasonable.
3) The results of TimeSeg seems to be great. The selection of metrics rae pretty straightforward and standard.

**Weaknesses:**

1) There are some parts related to RL towards the end of section 4 that may not be clear to non-RL-experts, like me. The overall summary of the method is not present in the main text. (Sections in the Appendix provides some details make it easier to understand.) Details are outlined in the "Questions" box below.
2) (This is not necessarily a weakness) While the premise of the paper is good, as shown in the abstract, "existing explanations focus on point-wise explainations, we introduce segment-wise explanations.", there is a lack of "interpreting" that segment - the method on "highlight" the segment. The paper introduces an method of explanation, with its output similar to previous approach. Thus this paper does not aim for higher level interpretations of the predictions. This means that the paper is not really "ground-breaking".

**Questions:**

As I am not familiar with RL at all, the questions I have may be a little basic. I would like to first summarize my understanding of the method (outlined in Section 4 and the Appendix)

I. Our goal is to find segments $s_{1:K}$ so that they "explain" the predictions. To do this, we maximize MI

II. Decompose MI in CMI and get equation 3. We can optimize equation 3 using RL framework and Problem is to optimize eq 3 in a RL framework using equation 5, with the reward in equation 4. Problem the sampling step is $\pi_\phi$ is not differentiable. Also $\pi_\phi$ are segments are therefore we only need to provide the "start" and the "end" for each segment. Thus $\pi_\phi$ can be decomposed into $\pi_{\phi^s}$ and $\pi_{\phi_e}$.

III. We can directly compute the gradient w.r.t $\phi$ and optimize (5), but the gradient variance is high. Thus, we need another loss term, equation (25) to reduces gradient variances, by introducing another network $V_\psi$ to estimate expected cumulative return.

IV. However, the new policy can diverge far away. So we need another loss term equation (26) instead. The total loss is a combination of (5), (25) and (26).

V. Now we have 2 networks, $\pi_\phi$ and $\pi_\psi$. It's time to train these networks. We jointly optimize these 2 during the training stage (as I understand because each loss has an expectation on x, and the number of steps are described in Appendix A2.)

VI. To generate explanations using inference data, we fix the 2 networks as they are trained. We then use equation 8) for a stopping criteria on whether new segments should be added.

For clarification, I wonder if the above description are correct or not.

Questions.
Q1. At inference time (if there is a difference between training vs inference), how do we select the next segment? Is it just sample from $\pi_\phi$ once? Or do we just do argmax? Or do we sample several times to see what which segment yields the highest reward?

Q2. As mentioned in L147, the segments are non-overlapping. But in the method section, there is no such constraints (in case I did not miss it). The only constraints are $t^s <= t^e$. So do we combine non overlapping segments?

Q3. I think A big part of Section 4.4 can be shortened, as it just describes a very trivial mechanism. The total loss, the training procedure and inference procedure can be described more clearly in the method section instead. (In case there is a separation between training and inference time)

Q4. (In case there is a separation between training and inference time,) how do we validate when the training is done? Do we validate on a separate validation set?

Q5. How does it apply to multivariate time series? Are there some rough run-time numbers?

Minor
1. The "s" is reused and it is slightly confusing. It can mean the segment index, or the "start" of the segment. It should not create confusion if the user is reading this carefully though. Changing this is optional.
2. In Table 2, what is the rationale of the highlighting? For example, Epilepsy, WinIT has 1.99 suff. while TimeSeg has 1.94 suff.. If we only look at the mean, TimeSeg should be bolded and WinIT should be underlined.
3. In Table 3, the errors should be subscripted (as in Table 2), for consistency.

---

> ### Author Response · Authors · 2025-11-21
>
> We sincerely thank reviewer **EPf4** for the thorough and thoughtful review. The reviewer's careful engagement with both the methodological formulation and practical aspects of our method has been invaluable in strengthening our manuscript.
> Based on the reviewer's feedback, we have:
>
> - Updated the manuscript to detail the value loss and the segment selection process at training and inference time (**Sec. 4**)
> - Further highlighted the interpretability aspect of the selected segments through clustering analysis (**Appendix D.8**)
> - Extended our work to multivariate analysis (**Appendix D.5**) and provided additional runtime analysis (**Appendix D.7**)
>
> We provide a point-by-point response below.

---

> > ### Author Response · Authors · 2025-11-21
> >
> > **1. Summarization of our work**
> >
> > Thank you for your thoughtful engagement with our work. We are pleased to see that your understanding aligns well with our methodology.
> > Here, we respectfully offer two minor clarifications:
> >
> > 1. Regarding (II): We maximize the joint MI (Eq. 1) and rewrite this equation via the chain rule as a sum of CMIs (Eq. 2).
> > We then make each CMI term tractable using a parameterized surrogate (Eq. 3).
> > Subsequently, we adopt an RL formulation where the per-step reward is the CMI surrogate (Eq. 4), and optimize the regularized expected return with sparsity and termination constraints (Eq. 5), which represents the learnable RL form of Eq. 2.
> >
> > 2. Regarding (VI): At inference time, we use only the policy network. The value network is only used exclusively during training for variance reduction. At inference, given an input x, we sequentially select informative segments through an argmax operation, stopping based on the termination criteria (Eq. 8).
> >
> > Your understanding of the remaining aspects is correct.
> > Based on your feedback, we have updated the main manuscript(**Sec 4.3, 4.4**) to provide further details clarifying the differences between training and inference procedures

---

> > > ### Author Response · Authors · 2025-11-21
> > >
> > > **[Q3] RL formulation in Sec 4**
> > >
> > > 1. Regarding Sec 4.3, we would like to to briefly note that the use of value function as an advantage term is a commonly used technique in RL, and not part of our methodological contribution, which is why the value-loss term was originally omitted from the main text.
> > >
> > > 2. Regarding Sec 4.4, we respectfully note that this section provides important implementation details of our work, including how masking is performed and how segments are constructed and fed into the explainer.
> > >
> > > However, based on the reviewer's feedback, we have included the value-loss term explicit in Sec. 4.3 and have enhanced Sec. 4.4 with additional details on the segment selection process at both training and inference time, to better clarify the overall training procedure and the segment selection process of our approach.

---

> > > > ### Author Response · Authors · 2025-11-21
> > > >
> > > > **[W2] Highlighting the Interpretability of Segments**
> > > >
> > > > We highly agree with the reviewer that discovering segments is not the end goal—the found segments should provide meaningful interpretability. We emphasize two key points regarding the interpretability of our segments:
> > > >
> > > > **1. Complete segment extraction:** TimeSeg outputs discrete, complete segments rather than point-wise or fragmented saliency scores. This enables downstream analysis that requires contiguous subsequences as interpretable units.
> > > >
> > > > **2. Enabling higher-level analysis:** The extracted segments can be used for further interpretable analysis of time series patterns. For example, we performed segment clustering analysis and visualized the cluster centroids (**Figure 9, Appendix D.8**). Such analysis is straightforward with our method but difficult with point-wise saliency approaches, which output only individual point values rather than coherent temporal patterns.

---

> > > > > ### Author Response · Authors · 2025-11-21
> > > > >
> > > > > **[Q5] Extension to Multivariate Settings**
> > > > >
> > > > > **Why Univariate?**
> > > > > We would first like to explain why we focused on a univariate setup. Here, our focus was to extract meaningful segments within a univariate data, without mixing in the separate problem of selecting relevant channels. As channels may have different importance for black-box model performance, analyzing meaningful segments in multivariate settings would require additional considerations for channel selection, which is orthogonal to our core contribution.
> > > > >
> > > > > However, we conducted additional experiments to demonstrate TimeSeg's applicability to multivariate time series as requested by the reviewer.
> > > > >
> > > > > **A Simple Extension to Multivariate**
> > > > >
> > > > > Here, we add a categorical channel action and define each segment as a joint draw over (channel, start, end), $\mathbf{s} = (c, t^s, t^e), \text{where }c \in [d]$. The black-box and objective remain the same; only the action space is augmented. Formally,
> > > > >
> > > > > $\pi_\phi(\mathbf{s} \mid \cdot)
> > > > > = \pi_\phi(c, t^s, t^e\mid\cdot)=\pi_{\phi^{\mathrm{c}}}(c \mid \cdot)\,
> > > > >   \pi_{\phi^{\mathrm{s}}}(t^{\mathrm{s}} \mid c, \cdot)\,
> > > > >   \pi_{\phi^{\mathrm{e}}}(t^{\mathrm{e}} \mid c, t^{\mathrm{s}},\cdot)$
> > > > >
> > > > > where, $c$ denotes the selected channel index, and $\pi_{\phi^{\mathrm{c}}}$ is a categorical distribution over channels whose logits are produced by the same backbone used in TimeSeg, i.e., a neural network parameterized by $\phi^{\mathrm{c}}$ that outputs channel selection probabilities.
> > > > >
> > > > > **Results on SeqComb-Multivariate (MV)**
> > > > >
> > > > > On SeqComb-MV with sparsity 0.06, our multivariate extension of TimeSeg achieves strong F1/IoU performance compared to baseline methods (TimeX++), with a contiguity score of only 0.003 (lower is better) compared to 0.064 for TimeX++. This demonstrates that TimeSeg identifies continuous, meaningful segments that are important for model performance, while maintaining better segment contiguity than existing methods. We provide the full results in **Appendix. D.5, Page. 22** of the updated manuscript.
> > > > >
> > > > > | Methods    | F1            | IoU           | Contiguity     |
> > > > > |------------|---------------|---------------|----------------|
> > > > > | TimeX++    | 0.471±0.038   | 0.334±0.030   | 0.064±0.009 |
> > > > > | Ours       | **0.502±0.034** | **0.353±0.032**   | **0.003±0.000** |

---

> > > > > > ### Author Response · Authors · 2025-11-21
> > > > > >
> > > > > > **[Q5] Runtime Numbers**
> > > > > >
> > > > > > We report the training wall-clock time per batch (batch size 256), averaged over 1,000 batches and inference wall-clock per samples averaged over 1000 samples for each method (**Table. 19**). For our method, the reported training time explicitly includes both the segment-selection rollout (policy interaction with the black-box) and the PPO update steps. LIMESegment is training-free, so only the inference time is reported. We show the computation time is highly comparable to baseline methods. In this setup, a full training run of TimeSeg takes approximately 20–30 minutes.
> > > > > >
> > > > > > | **Method** | **Rollout Time (s)** | **Training Time (s)**  | **Inference Time (s)** |
> > > > > > | --- | --- | --- | --- |
> > > > > > | **LIMEsegment** | – | – | 6.166±1.620 |
> > > > > > | **TimeX++** | – | 0.250±0.025 | 0.003±0.000 |
> > > > > > | **Ours** | 0.267±0.059 | 0.040±0.075 | 0.023±0.036 |

---

> > > > > > > ### Author Response · Authors · 2025-11-21
> > > > > > >
> > > > > > > **Other Details**
> > > > > > >
> > > > > > > **[Q1] Selection at Inference time**
> > > > > > > At infrerence time, we do not search for the “highest-reward” segment as the reward is used only for training.
> > > > > > > Instead, at inference, we use the learned (amortized) policy to output a deterministic argmax at each step until Eq. (8) holds (no multi-sample probing).
> > > > > > >
> > > > > > > **[Q2] Overlapping Segments**
> > > > > > > The final explanations are non-overlapping by construction, as segments are merged into a single gate if they contain intervals.
> > > > > > > Also, overlapping is penalized at selection — the union length increases with limited marginal CMI gain and the length penalty (controlled by $\lambda$) penalizes redundancy.
> > > > > > >
> > > > > > > **[Q3] Use of Validation**
> > > > > > > We use a separate validation set and select the best epoch by validation reward (early stopping on the highest validation reward).
> > > > > > >
> > > > > > > **Minors**
> > > > > > > [1] Thank you for the valuable feedback on notation. We considered alternative notations but found that changes might introduce additional confusion. We have therefore retained the current notation for consistency.
> > > > > > > [2] Yes, the reviewer is correct. This was an error, and we have corrected it in the updated manuscript. Thank you for catching this!
> > > > > > > [3] We have made the suggested changes based on your feedback.

---

> > > > > > > > ### Author Response · Authors · 2025-11-21
> > > > > > > >
> > > > > > > > We sincerely thank the reviewer once again for the constructive feedback that has strengthened our work.
> > > > > > > > We hope this response addresses your questions, and we remain available for any further clarification or discussion.

---

> > > > > > > > > ### Comment · Reviewer_EPf4 · 2025-11-24
> > > > > > > > >
> > > > > > > > > Thanks for the response!
> > > > > > > > >
> > > > > > > > > **[Summary]** Thanks for the clarification of the work.
> > > > > > > > >
> > > > > > > > > **[Q3]** Thanks for including the value loss term (and also thanks for not updating the equation number for better comparison). But after the response period ends, the author should make that equation numbered as well. For section 4.4, I was suggesting shortening it because, as you mentioned, it is about implementation, instead of theory or method. But in case the paper can fit into the required number of pages while adding more clarification in 4.3, it is okay to include that.
> > > > > > > > >
> > > > > > > > > **[W2]** Thanks for the response. The inclusion of the further interpretability analysis is nice.
> > > > > > > > >
> > > > > > > > > **[Q5]** So in summary, we are extracting windows per channel separately. Thanks for the experiments. Thanks for the runtime numbers as well.
> > > > > > > > >
> > > > > > > > > In summary, I am happy with the response. Thanks a lot.

---

> ### Author Response · Authors · 2025-11-25
>
> Dear Reviewer **EPf4**
>
> We are sincerely grateful for your time and helpful feedback in the review process, which has greatly helped us improve our work. We are delighted to see that the reviewer has found our response satisfiable. In light of your satisfaction with our response, we wonder whether the reviewer would kindly consider revising the rating.
>
> Best regards,
>
> Authors

---

### Official Review · Reviewer_MVJy · 2025-10-29

**Soundness:** 3
**Presentation:** 4
**Contribution:** 3
**Rating:** 6
**Confidence:** 5

**Summary:**

The paper presents TimeSeg, an information-theoretic framework that explains black-box time-series models by selecting contiguous temporal segments that best capture information relevant to predictions. Using reinforcement learning, it sequentially identifies predictive segments for each instance. This approach provides interpretable, segment-wise explanations that reveal holistic, class-predictive temporal patterns.

**Strengths:**

The paper addresses an interesting problem with a novel and well-conceived approach. It is clearly written and well-organized. The experiments are comprehensive, covering diverse datasets and evaluation metrics.

**Weaknesses:**

The paper overlooks relevant prior work. Specifically, it omits discussion of Theissler, A., et al. (2022). Explainable AI for time series classification: a review, taxonomy and research directions. IEEE Access, 10, 100700–100724, which provides a comprehensive survey on explainable AI for time series. It also fails to reference Spinnato, F., et al. (2023). Understanding any time series classifier with a subsequence-based explainer. ACM Transactions on Knowledge Discovery from Data, 18(2), 1–34, which presents a directly related method. Furthermore, the authors set τ = 0.3 and Kmax = 5 without explaining the rationale or analyzing the sensitivity of results to these parameters. Finally, the experiments are restricted to a single black-box model (CNN), overlooking state-of-the-art time-series classifiers such as ROCKET (Dempster, A., et. al (2020). ROCKET: exceptionally fast and accurate time series classification using random convolutional kernels. Data Mining and Knowledge Discovery, 34(5), 1454-1495.), MiniROCKET (Dempster, A., et. al.. (2021, August). Minirocket: A very fast (almost) deterministic transform for time series classification. In Proceedings of the 27th ACM SIGKDD conference on knowledge discovery & data mining (pp. 248-257).), and MR-HYDRA (Dempster, A., et. al. (2023). Hydra: Competing convolutional kernels for fast and accurate time series classification. Data Mining and Knowledge Discovery, 37(5), 1779-1805.). The evaluation should include multiple black-box models to better demonstrate the robustness and model-agnostic nature of the proposed approach.

**Questions:**

Questions can be derived from the above weaknesses.

---

> ### Author Response · Authors · 2025-11-21
>
> We thank the reviewer **MVJy** for finding our experiments comprehensive, and providing detailed feedback to strengthen our positioning. We appreciate the opportunity to further strengthen the work based on the reviewer’s feedback.
>
> Our response is organized into three key parts:
>
> **1. Positioning of our work:** We provide detailed explanation on how TimeSeg differs with **Theissler et al. (2022)** and **Spinnato et al. (2023/2024)** in both problem formulation and objectives.
>
> **2. Sensitivity Analysis on Key Hyperparmeters:** We perform a sensitivity analysis over $\tau$ and $K_{\max}$, assessing their effect on sparsity, segment count, and performance.
>
> **3. Blackbox Generality:** Based on the reviewer's suggestion, we perform our experiment on  **MiniROCKET** as a blackbox, as well as **RNN** and **Transformer** models.

---

> ### Author Response · Authors · 2025-11-21
>
> **[W1] 1. Positioning of our work:**
>
> **Different explanation focus of TimeSeg**
>
> We highlight that LASTS is a **shapelet-based global explainer** that provides instance-wise explanations using fixed-length global shapelets. In contrast, **TimeSeg provides instance-wise feature attribution**, identifying which segments were important for each individual time series prediction.
>
> The key distinction is the question being answered: TimeSeg addresses **"For this particular input, which contiguous subsequences were responsible for its prediction?"** rather than constructing a global library of prototypical patterns that explain the classifier as a whole. The former provides local explanations while the latter offers global, model-approximation explanations. This **positions TimeSeg on a fundamentally different methodological axis from shapelet approaches like LASTS**, which are better viewed as global, model-approximation methods.

---

> ### Author Response · Authors · 2025-11-21
>
> **[W2] 2. Sensitivity Analysis on $\tau, K_{\max}$ :**
>
> As the reviewer suggested, we perform an additional analysis on the hyperparameters $\tau, K_{\max}$.
> Here, $\tau$ controls when the policy should stop adding new segments, while $K_{\max}$ is the maximum number of segments that can be added.
> We note that $K_{\max}$ serves primarily as a safety constraint to prevent unbounded segment generation; in practice, a well-trained policy typically terminates well before reaching this limit.
> The actual number and length of segments are governed mainly by $\tau$ (termination threshold) and $\lambda$ (sparsity regularization).
> We add a part of full sensitivity analysis results for $K_{\max}$ and $\tau$ below, and the full version is reported in **Tables 10 and 11 of the updated manuscript**.
>
>
> ### Ablation on $\tau$
> | $\tau$ | Sparsity | F1 | N segments | Each segment Length |
> | --- | --- | --- | --- | --- |
> | -0.1 | 0.170±0.037 | 0.523±0.031 | 4.747±0.038 | 10.219±2.280 |
> | 0 | 0.193±0.029 | 0.627±0.026 | 3.785±0.059 | 16.998±2.186 |
> | 0.1 | 0.143±0.021 | 0.667±0.024 | 1.865±0.057 | 16.607±2.509 |
> | 0.3 | 0.141±0.016 | 0.645±0.018 | 1.631±0.060 | 17.615±2.869 |
> | 0.5 | 0.137±0.030 | 0.618±0.012 | 1.521±0.040 | 16.364±2.265 |
> | 0.9 | 0.136±0.020 | 0.599±0.013 | 1.300±0.031 | 19.487±2.370 |
>
> ### Ablation on $K_{\max}$
> | $K_{\max}$ | Sparsity | F1 | N segments | Each segment Length |
> | --- | --- | --- | --- | --- |
> | 1 | 0.188±0.017 | 0.499±0.013 | 1.000±0.000 | 37.549±3.473 |
> | 2 | 0.170±0.022 | 0.643±0.024 | 1.619±0.056 | 21.751±3.416 |
> | 3 | 0.154±0.017 | 0.652±0.011 | 1.647±0.060 | 18.102±2.412 |
> | 5 | 0.141±0.016 | 0.645±0.018 | 1.631±0.060 | 17.615±2.869 |
> | 7 | 0.140±0.025 | 0.642±0.015 | 1.647±0.061 | 17.610±3.789 |
>
> The results show that $\tau$ mainly trades off the number vs. length of segments within the sparsity budget, and that once $K_{\max} \ge 3$, performance becomes stable and the bound is rarely reached.

---

> ### Author Response · Authors · 2025-11-21
>
> **[W3] 3. Blackbox Generality (MiniRocket, RNN, Transformer)**
>
> Based on the reviewer's feedback, we performed additional experiments with different black-box architectures, including MiniRocket, RNN, and Transformer, with a summary table below and full results reported in **Figure 8** and **Tables 14-18**.
> Notably, MiniRocket uses a fixed transform plus linear head structure that does not expose gradients or usable embeddings. Consequently, **gradient- and embedding-based explainers (e.g., IG, TimeX++) are not applicable to such architectures, whereas TimeSeg remains fully functional as it operates purely through black-box outputs.**
>
> In the below we share the results on MiniRocket.
>
> SeqCombSinlge with sparsity 0.65
>
> | Model | F1 | IoU | Continuity |
> | --- | --- | --- | --- |
> | WinIT | 0.306±0.050 | **0.247±0.042** | 0.044±0.011 |
> | Limesegment | 0.314±0.046 | 0.189±0.034 | 0.010±0.004 |
> | Ours | **0.378±0.078** | 0.241±0.065 | **0.010±0.001** |
>
> GunPoint with sparsity 0.54
>
> | Model | Suff | Comp | Continuity |
> | --- | --- | --- | --- |
> | Random | 20.400±6.896 | 23.590±7.752 | 0.502±0.002 |
> | WinIT | 0.018±0.040 | 51.734±16.855 | 0.029±0.003 |
> | Limesegment | 1.067±1.406 | 62.002±5.895 | 0.020±0.001 |
> | Ours | **0.014±0.032** | **92.990±7.861** | **0.017±0.001** |
>
> Since MiniROCKET’s linear head relies on many randomly convolved features spread across the whole signal rather than a few localized regions, TimeSeg naturally selects more segments (i.e., higher sparsity) to approximate its behavior faithfully.
>
> Also, we share summary tables on SeqCombSingle, MIT-ECG, and GunPoint.
>
> ### SeqCombSingle — F1
> | Backbone | LIMESegment | TimeX++ | TimeSeg (Ours) |
> | --- | --- | --- | --- |
> | RNN | 0.432 ± 0.013 | **0.601 ± 0.079** | 0.595 ± 0.011 |
> | Transformer | 0.338 ± 0.013 | 0.535 ± 0.133 | **0.573 ± 0.014** |
>
> ### MIT-ECG — F1
> | Backbone | LIMESegment | TimeX++ | TimeSeg (Ours) |
> | --- | --- | --- | --- |
> | RNN | 0.608 ± 0.015 | 0.582 ± 0.123 | **0.746 ± 0.022** |
> | Transformer | 0.461 ± 0.019 | 0.572 ± 0.132 | **0.611 ± 0.040** |
>
> ### GunPoint — Sufficiency
> | Backbone | LIMESegment | TimeX++ | TimeSeg (Ours) |
> | --- | --- | --- | --- |
> | RNN | 3.049 ± 1.533 | 36.322 ± 1.555 | **1.557 ± 0.903** |
> | Transformer | 22.703 ± 15.732 | 44.995 ± 8.446 | **-0.563 ± 2.452** |
>
> These results show that TimeSeg consistently performs well across diverse black-box architectures (including MiniROCKET, RNN, and Transformers), even without accessing gradients or embeddings, demonstrating strong practical utility in fully strict black-box settings.

---

> ### Author Response · Authors · 2025-11-21
>
> We sincerely appreciate the reviewer's constructive feedback, which has enabled us to conduct more comprehensive analysis and strengthen our results.
> We remain available for any further discussion or clarification as needed.

---

### Official Review · Reviewer_jJZe · 2025-10-31

**Soundness:** 2
**Presentation:** 3
**Contribution:** 3
**Rating:** 4
**Confidence:** 4

**Summary:**

The paper proposes an information-theoretic framework named TimeSeg for segment-wise explanation of black-box time-series models. The core motivation stems from the limitation of existing approaches, which mostly provide point-wise or fixed-patch explanations and lack a principled definition of meaningful temporal segments. TimeSeg aims to identify a set of continuous temporal segments that maximize the mutual information with the model’s prediction outcomes. The authors formalize this as a segment selection problem that seeks to maximize the joint mutual information between the selected segments and the predictive output. Since direct optimization is intractable, the problem is reformulated under a reinforcement learning framework to approximate the optimal segmentation policy. Specifically, a policy network predicts the start and end indices of informative segments and applies a sparsity regularization to control the explanation length. The policy is trained using Proximal Policy Optimization to ensure stable learning. Experiments conducted on multiple synthetic and real-world time-series datasets demonstrate that the proposed method produces more human-aligned and coherent explanations while preserving predictive performance.

**Strengths:**

The paper provides a clear theoretical formulation, defining segment-wise explanation for black-box time-series models as an optimization problem that seeks continuous subsequences maximizing the joint mutual information with the target prediction.

The authors transform the sequential decision process of maximizing conditional mutual information into a reinforcement learning task, enabling efficient exploration of the optimal explanatory segments within a discrete and non-differentiable selection space.

The proposed method operates effectively under a strict black-box setting, without requiring access to internal gradients or model parameters, demonstrating strong practical applicability. Moreover, the generated explanatory segments are coherent and cognitively aligned with human reasoning, showing high interpretability and real-world utility.

**Weaknesses:**

The evaluation metrics adopted in the current experiments lack sufficient persuasive power and fail to comprehensively reflect the proposed method’s advantages and degree of improvement.

All experiments are conducted solely on univariate time-series datasets, leaving the method’s applicability and robustness in multivariate scenarios unverified.

Moreover, the evaluation is performed exclusively on the Temporal Convolutional Network , without validation on other black-box architectures such as RNNs or Transformers, which limits the evidence for the method’s generality.

In addition, the paper does not report the computational cost associated with the reinforcement learning process, nor does it provide a systematic analysis of the algorithm’s efficiency, stability, or convergence behavior. The reproducibility details are relatively fragmented, and key hyperparameters (e.g., PPO parameters, learning rate) are not clearly summarized in the main text, which hinders reproducibility and engineering-level assessment of the proposed approach.

**Questions:**

**On the Reinforcement Learning Training Process**

a. Reinforcement learning typically requires extensive interactions between the agent and the environment (i.e., the black-box model) to sample training data. However, the paper does not report the computational cost of training and inference, nor does it provide an analysis of the algorithm’s efficiency or convergence.

b. The paper sets the maximum number of segments to $K_{\text{max}} = 5$. Could this constraint limit the number of agent–environment interactions and thus affect training performance? Please include a sensitivity analysis regarding this hyperparameter.

c. Please clarify whether the reinforcement learning training exhibits stability and consistency under different random seeds.

**On Experimental Evaluation and Comparisons**

a. Although the paper compares against methods such as TimeX++, the datasets and evaluation metrics differ from those used in the original TimeX++ experiments. Why not adopt the same evaluation metrics and experimental settings to ensure fair comparison? It is recommended to include results under consistent metrics such as AUPRC, AUP, and AUR to enhance persuasiveness.

b. All experiments are conducted exclusively on the TCN architecture. Please further validate the explainability and generalization of TimeSeg on other black-box models such as RNNs and Transformers.

c. The paper does not include TimeX as a baseline, even though it represents a key prior work in time-series explainability. The absence of this comparison reduces the completeness and competitiveness of the experimental conclusions.

**On Methodological Applicability**

The proposed method is evaluated only on univariate time-series datasets. Please elaborate on the framework’s applicability and potential extensions to multivariate time-series scenarios.

---

> ### Author Response · Authors · 2025-11-21
>
> We sincerely thank reviewer **jJZe** for finding our theoretical formulation clear and recognizing the value of our work in real-world setups. We appreciate the opportunity to improve our work based on the reviewer's feedback.
> Below, we have prepared a detailed response to address the questions raised by the reviewer.
>
> Our response is organized into four key parts:
>
> **1. Justification of the Metrics:** We justify that the metric used was the most appropriate for segment-wise explanations such as those provied by our method.
>
> **2. Extension to Multivariate Settings:** While our work focuses on a univariate setup (L130-131), we perform an additional experiment extending our method to a multivariate setup as requested by the reviewer.
>
> **3. Additional Blackbox Backbones:** We perform additional experiments with RNN and Transformer backbones as noted by the reviewer.
>
> **4. Hyperparameters, Compute Time, Analysis of Convergence, Sensitivity Analysis, Random Seeds:** We report the full hyperparameters used, provide analysis of convergence, sensitivity analysis and report computation time.
> Full results are reported in Appendix C, D of the manuscript.

---

> ### Author Response · Authors · 2025-11-21
>
> **[W1, Q2-a] 1. Justification of the Metrics**
>
> **The objective of segment-wise explanation and the need for F1/IoU.**
>
> Our explainer outputs a **discrete set of contiguous segments** represented as binary gates. For instance, our explanation takes the form [0, 0, 1, 1, 1, 0, 0, 0], where contiguous sequences of 1s represent segments identified as important. Importantly, this is not a continuous per-timestep score like [0.1, 0.2, 0.5, 0.7, 0.6, 0.3, 0.1].
> For segment-wise, binary selections with ground-truth masks, the evaluation objective should be: **"Do the selected regions match the ground truth?"** We therefore report F1/IoU (overlap metrics).
> Here, both F1 and IoU effectively summarize precision and recall for binary masks, making them the most suitable measures for evaluating segment-wise explanations.
> However, AUPRC/AUP/AUR used in the previous point-wise methods assumes the importance scores to be continuous and our explainer returns a discrete union of segments without continuous importance scores.
> Consequently, metrics based on areas under curves are unsuitable for evaluating segment-wise explanations.

---

> ### Author Response · Authors · 2025-11-21
>
> **[W2] 2. Extension to Multivariate Settings**
>
> **Why Univariate?**
>
> We purposely focused on finding meaningful segments within univariate data, without mixing in the separate problem of selecting relevant channels. As channels may have different importance for black-box model performance, analyzing meaningful segments in multivariate settings would require additional considerations for channel selection, which is orthogonal to our core contribution. Nevertheless, **we have conducted additional experiments to demonstrate TimeSeg's applicability to multivariate time series as requested by the reviewer**, with results reported in Appendix. D.5, Page. 22.
>
> **A Simple Extension to Multivariate**
>
> We add a categorical channel action and define each segment as a joint draw over (channel, start, end), $\mathbf{s} = (c, t^s, t^e), \text{where }c \in [d]$. The black-box and objective remain the same; only the action space is augmented. Formally,
>
> $\pi_\phi(\mathbf{s} \mid \cdot)
> = \pi_\phi(c, t^s, t^e\mid\cdot)=\pi_{\phi^{\mathrm{c}}}(c \mid \cdot)\,
>   \pi_{\phi^{\mathrm{s}}}(t^{\mathrm{s}} \mid c, \cdot)\,
>   \pi_{\phi^{\mathrm{e}}}(t^{\mathrm{e}} \mid c, t^{\mathrm{s}},\cdot)$
>
> Here, $c$ denotes the selected channel index, and $\pi_{\phi^{\mathrm{c}}}$ is a categorical distribution over channels whose logits are produced by the same backbone used in TimeSeg, i.e., a neural network parameterized by $\phi^{\mathrm{c}}$ that outputs channel selection probabilities.
>
> **Results on SeqComb-Multivariate (MV)**
>
> On SeqComb-MV with sparsity 0.06, our multivariate extension of TimeSeg achieves strong F1/IoU performance compared to baseline methods (TimeX++), with a contiguity score of only 0.003 (lower is better) compared to 0.064 for TimeX++. This demonstrates that TimeSeg identifies continuous, meaningful segments that are important for model performance, while maintaining significantly better segment contiguity than existing methods. We provide the full results in Appendix. D.5, Page. 22 of the updated manuscript.
>
> | Methods | F1 | IoU | Contiguity |
> | --- | --- | --- | --- |
> | TimeX++ | 0.471±0.038 | 0.334±0.030 | 0.064±0.009 |
> | Ours | **0.502±0.034** | **0.353±0.032** | **0.003±0.000** |

---

> ### Author Response · Authors · 2025-11-21
>
> **[W3, Q2-b] 3. Additional Blackbox Backbones**
>
> Beyond TCN, we evaluate TimeSeg on RNN and Transformer black-box models across three dataset types: SeqCombSingle (synthetic), MIT-ECG (real-world with ground-truth labels), and GunPoint (real-world without ground-truth). TimeSeg demonstrates consistent and stable performance across different architectures while remaining superior to baseline methods (e.g., LimeSegment, TimeX++). We provide a summary table below and figure, in **Figure 7 Page 23**, with full results reported in **Appendix. D.6, Page. 23** of the updated manuscript.
>
> SeqCombSingle — F1
> | Backbone | LIMESegment | TimeX++ | TimeSeg (Ours) |
> | --- | --- | --- | --- |
> | RNN | 0.432 ± 0.013 | **0.601 ± 0.079** | 0.595 ± 0.011 |
> | Transformer | 0.338 ± 0.013 | 0.535 ± 0.133 | **0.573 ± 0.014** |
>
> MIT-ECG — F1
> | Backbone | LIMESegment | TimeX++ | TimeSeg (Ours) |
> | --- | --- | --- | --- |
> | RNN | 0.608 ± 0.015 | 0.582 ± 0.123 | **0.746 ± 0.022** |
> | Transformer | 0.461 ± 0.019 | 0.572 ± 0.132 | **0.611 ± 0.040** |
>
> GunPoint — Sufficiency
> | Backbone | LIMESegment | TimeX++ | TimeSeg (Ours) |
> | --- | --- | --- | --- |
> | RNN | 3.049 ± 1.533 | 36.322 ± 1.555 | **1.557 ± 0.903** |
> | Transformer | 22.703 ± 15.732 | 44.995 ± 8.446 | **-0.563 ± 2.452** |
>
> These results show that TimeSeg consistently performs well across diverse black-box architectures (TCN, RNN, Transformer), maintaining strong segment-wise explanations regardless of the underlying model.

---

> ### Author Response · Authors · 2025-11-21
>
> **4. Hyperparameters, Compute Time, Analysis of Convergence, Sensitivity Analysis, Random Seeds**
>
> **[Q1-a] Hyperparameters**
>
> We provide the full hyperparameters (buffer/rollout/batch, PPO settings, lrs/scheduler, sparsity/termination, policy configs) used in our experiment and have updated our manuscript accordingly **Table 7**.  We are commited to make our work fully reproducible, and have also made the codes publicly available.
>
> **[Q1-a] Compute Time**
>
> We report the training wall-clock time per batch (batch size 256), averaged over 1,000 batches and inference wall-clock per samples averaged over 1000 samples for each method (see **Table 19**). For our method, the reported training time explicitly includes both the segment-selection rollout (policy interaction with the black-box) and the PPO update steps. LIMESegment is training-free, so only the inference time is reported. We show the computation time is highly comparable to baseline methods. In this setup, a full training run of TimeSeg takes approximately 20–30 minutes.
>
> | **Method** | **Rollout Time (s)** | **Training Time (s)**  | **Inference Time (s)** |
> | --- | --- | --- | --- |
> | **LIMEsegment** | – | – | 6.166±1.620 |
> | **TimeX++** | – | 0.250±0.025 | 0.003±0.000 |
> | **Ours** | 0.267±0.059 | 0.040±0.075 | 0.023±0.036 |
>
> **[Q1-a] Convergence Analysis**
>
> We report the training/validation total reward curves in **Figure 8, Page. 26** of the updated manuscript. We also provide a separate reward curve for cross-entropy and length (sparsity) to illustrate how each regularizer shapes the learning process.
> **In summary, our model leads to stable convergence upon training.**
>
>
> **[Q1-a] Interaction Cost**
>
> Standard RL often runs ~10³–10⁴ steps per episode, yielding heavy interaction costs. We cast RL as selection: per sample we take at most $K_{\max}$ decisions (small), and cap per-epoch sampling (e.g., $N_{\text{rollout}}=1024$). Trajectories go to a buffer and are reused until eviction. **Interaction cost is thus far below standard RL and comparable to STE-based masking**.

---

> ### Author Response · Authors · 2025-11-21
>
> **[Q1-b] Sensitivity Analysis on $\tau, K_{\max}$**
>
> $\tau$ controls when the policy should stop adding new segments, while $K_{\max}$ is the maximum number of segments. Here, $K_{\max}$ is a hard upper bound, the number of segment rarely reach it with a moderately large value. The actual number and length of segments are governed mainly by $\tau$ (termination) and $\lambda$ (sparsity). We add a part of full sensitivity analysis results for $K_{\max}$ and $\tau$ below and the full version is reported in **Tables 10 and 11 of the updated manuscript**.
>
> ### Ablation on $\tau$
> | $\tau$ | Sparsity | F1 | N segments | Each segment Length |
> | --- | --- | --- | --- | --- |
> | -0.1 | 0.170±0.037 | 0.523±0.031 | 4.747±0.038 | 10.219±2.280 |
> | 0 | 0.193±0.029 | 0.627±0.026 | 3.785±0.059 | 16.998±2.186 |
> | 0.1 | 0.143±0.021 | 0.667±0.024 | 1.865±0.057 | 16.607±2.509 |
> | 0.3 | 0.141±0.016 | 0.645±0.018 | 1.631±0.060 | 17.615±2.869 |
> | 0.5 | 0.137±0.030 | 0.618±0.012 | 1.521±0.040 | 16.364±2.265 |
> | 0.9 | 0.136±0.020 | 0.599±0.013 | 1.300±0.031 | 19.487±2.370 |
>
> ### Ablation on $K_{\max}$
> | $K_{\max}$ | Sparsity | F1 | N segments | Each segment Length |
> | --- | --- | --- | --- | --- |
> | 1 | 0.188±0.017 | 0.499±0.013 | 1.000±0.000 | 37.549±3.473 |
> | 2 | 0.170±0.022 | 0.643±0.024 | 1.619±0.056 | 21.751±3.416 |
> | 3 | 0.154±0.017 | 0.652±0.011 | 1.647±0.060 | 18.102±2.412 |
> | 5 | 0.141±0.016 | 0.645±0.018 | 1.631±0.060 | 17.615±2.869 |
> | 7 | 0.140±0.025 | 0.642±0.015 | 1.647±0.061 | 17.610±3.789 |
>
> The results show that $\tau$ mainly trades off the number vs. length of segments within the sparsity budget, and that once $K_{\max} \ge 3$, performance becomes stable and the bound is rarely reached.
>
>
> **[Q1-c] Multiple random seeds**
> While we have reported the mean and standard deviation from five-fold validation (each with distinct runs), **we also report multi-seed results on a single-fold experiment as the reviewer requested**. We show that the standard deviation is similar to the five-fold result.
>
> | Datasets | F1 | IoU | Contiguity | Sparsity |
> | --- | --- | --- | --- | --- |
> | SeqCombSingle | 0.651±0.011 | 0.510±0.014 | 0.017±0.000 | 0.142±0.031 |
> | LowVarDetectSingle | 0.496±0.009 | 0.363±0.009 | 0.020±0.000 | 0.174±0.015 |
> | FreqShapeVar | 0.724±0.022 | 0.567±0.020 | 0.073±0.002 | 0.099±0.005 |
> | MITECG | 0.736±0.021 | 0.617±0.028 | 0.006±0.000 | 0.139±0.004 |
>
> **[Q2-C] TimeX Results.**
>
> TimeX++ is asuccessor and improvement over TimeX. In our experiments, TimeX++ consistently outperformed TimeX on our benchmarks, so we chose TimeX++ as the primary baseline in the main paper. For completeness, we provide a table with TimeX results below.
>
> | Datasets | F1 (TimeX) | F1 (TimeX++)  | IoU (TimeX) | IoU (TimeX++) | Contiguity (TimeX) | Contiguity (TimeX++) |
> | --- | --- | --- | --- | --- | --- | --- |
> | SeqCombSingle | 0.618±0.061 | **0.636±0.157**  | 0.436±0.055 | **0.489±0.159**  | 0.118±0.020 | **0.098±0.015** |
> | LowVarDetectSingle | **0.442±0.043**  | 0.432±0.082 | **0.301±0.077** | 0.291±0.057 | 0.261±0.046 | **0.231±0.025** |
> | FreqShapeVar | 0.728±0.008 | **0.799±0.001** | 0.602±0.002 | **0.666±0.002** | 0.142±0.002 | **0.120±0.001** |
> | MITECG | 0.492±0.115 | **0.593±0.146** | 0.410±0.105 | **0.460±0.142** | 0.019±0.003 | **0.016±0.002** |

---

> ### Author Response · Authors · 2025-11-21
>
> We once again thank the reviewer for providing helpful feedback to improve our work.
> Please kindly let us know if there are any additional questions so that we can address them. Thank you.

---

> > ### Author Response · Authors · 2025-11-28
> >
> > Dear Reviewer **jJZe**,
> >
> > Thank you very much for your valuable comments and insightful suggestions on our manuscript. We are also excited to receive your encouraging feedback. We have carefully addressed the concerns raised in your review and revised our manuscript substantially. We have uploaded the revised manuscript, and all changes made to the manuscript are indicated in red throughout the revised main text and appendix. If there are any remaining issues or questions you would like us to address before the rebuttal deadline, please do not hesitate to let us know. We are more than happy to take the valuable opportunity to discuss our work with you further or make any further adjustments as needed.
> >
> > Thank you again for your time and positive feedback.
> >
> > Best,
> >
> > Authors

---

### Author Response · Authors · 2025-11-30
**General Response by Authors**

We sincerely thank reviewers `jJZe`, `MVJy`, `EPf4` for their thorough evaluation of our manuscript and their valuable feedback.

In this paper, we present **TimeSeg**, a *segment-wise explainer for black-box time-series models.* TimeSeg is a novel information-theoretic framework that employs reinforcement learning to sequentially identify predictive temporal segments at a per-sample level.

Overall, reviewers have found our work **novel and well-conceived, with clear theoretical formulation** (`jJZe`), supported by **comprehensive experimental validation** (`MVJy`), **appropriate evaluation metrics** (`MVJy`, `EPf4`), and **strong empirical performance demonstrating high interpretability and practical utility in time-series domains** (`jJZe`, `EPf4`).

We greatly appreciate these positive assessments and have made our best effort to address all questions raised by the reviewers to improve and clarify our work. Accordingly, we have carefully revised the manuscript with the following key additions:

- Experiments across **different black-box architectures** including RNN, Transformer, and MiniROCKET (Appendix D.6, Figure 8, Tables 14-18)  (`jJZe`, `MVJy`)
- **Sensitivity analysis** over hyperparameters $\tau$ and $K_{max}$ (Appendix D.4, Tables 10-11) (`jJZe`, `MVJy`)
- Extension to **multivariate settings** (Appendix D.5, Table 12) (`jJZe`, `EPf4`)
- Training and inference **computation time analysis** (Appendix D.7, Table 19), along with efficiency, stability, and convergence reports including reward curves (Appendix D.7, Figure 8) (`jJZe`, `EPf4`)
- **Interpretability analysis** with segment visualizations (Appendix D.9, Figure 9) (`EPf4`)
- Explicit **clarification of the value loss term** (Section 4.3) (`EPf4`)

All revisions are highlighted in red in the manuscript. We once again thank the reviewers for their insightful comments and constructive suggestions, which have significantly helped us improve our work. We remain available for any questions or discussions.

Note: Notably, reviewer `MVJy` had **increased the score from 6 to 8 during the rebuttal process**, but this change was lost due to the rollback.

Best,

Authors

---

### Meta-Review · Area_Chair_j6BV · 2026-01-06

**Summary:**

This paper received mixed review scores: 4, 6, 6. Reviewers overall recognized the merit of this work, regarding the proposed approach novel with clear theoretical formulation, the paper well written, and the results comprehensive. They also raised some concerns and suggestions, mainly about

1)	The selected evaluation metrics lack sufficient justification (jJZe);
2)	The applicability to multivariate time-series problems (jJZe, EPf4);
3)	Evaluation is mainly performed on Temporal Convolutional Network, missing comparison with other architectures such as RNNs and Transformers (jJZe, MVJy);
4)	Overlooked many relevant prior work (MVJy);
5)	Computational cost analysis and hyperparameter ablations (jJZe, MVJy, EPf4);
6)	Writing clarity about RL background for non-RL-experts (EPf4).

The authors provided detailed rebuttal to address the listed concerns point by point, with additional experimental results and analysis. Reviewer EPf4 explicitly acknowledged that the response is satisfactory. Other two reviewers didn’t have a chance to provide feedback. Instead, this AC carefully read the rebuttal and found it comprehensive and persuasive. Based on the review and rebuttal, this AC recommends accepting the paper.

**Reviewer Concerns:**

The rebuttal was very well prepared and addressed most reviewer conerns, especially for the first reviewer jJZe, who gave the score 4 and raised concerns about evaluation metrics, extension to multivariate time-series problems, and comparison with other architectures like Transformers.

I don't see any concerns that are still outstanding.

**Reviewer Scores:**

I think Reviewer jJZe would have changed his/her score to a more positive one, as the authors have addressed most of his/her concerns.

---

### Decision · Program_Chairs · 2026-01-26

Accept (Poster)